# Enhancing unity-based AR with optimal lossless compression for digital twin assets

**Mohammed Hlayel**[1,2]*, **Hairulnizam Mahdin**[2], **Mohammad Hayajneh**[3]*, **Saleh H. AlDaajeh**[4], **Siti Salwani Yaacob**[5], **Mazidah Mat Rejab**[2]

**1** Fatima College of Health Sciences, Institute of Applied Technology, Al-Ain, Abu Dhabi, UAE, **2** Faculty of Computer Science and Information Technology, Universiti Tun Hussein Onn Malaysia, Johor, Malaysia, **3** College of Information Technology, United Arab Emirates University, Al-Ain, Abu Dhabi, UAE, **4** Department of Information Systems & Security, College of Information Technology, United Arab Emirates University, Al-Ain, Abu Dhabi, UAE, **5** Faculty of Computing, Universiti Malaysia Pahang Al-Sultan Abdullah, Pekan, Pahang, Malaysia

\* mohammed.hlayelfchs.ac.ae (MH); mhayajnehuaeu.ac.ae (MH)

**Data Availability Statement:** https://github.com/Sahermatter2024/UnityAR-Compression-Comparison-Benchmark.

## Abstract

The rapid development of Digital Twin (DT) technology has underlined challenges in resource-constrained mobile devices, especially in the application of extended realities (XR), which includes Augmented Reality (AR) and Virtual Reality (VR). These challenges lead to computational inefficiencies that negatively impact user experience when dealing with sizeable 3D model assets. This article applies multiple lossless compression algorithms to improve the efficiency of digital twin asset delivery in Unity's AssetBundle and Addressable asset management frameworks. In this study, an optimal model will be obtained that reduces both bundle size and time required in visualization, simultaneously reducing CPU and RAM usage on mobile devices. This study has assessed compression methods, such as LZ4, LZMA, Brotli, Fast LZ, and 7-Zip, among others, for their influence on AR performance. This study also creates mathematical models for predicting resource utilization, like RAM and CPU time, required by AR mobile applications. Experimental results show a detailed comparison among these compression algorithms, which can give insights and help choose the best method according to the compression ratio, decompression speed, and resource usage. It finally leads to more efficient implementations of AR digital twins on resource-constrained mobile platforms with greater flexibility in development and a better end-user experience. Our results show that LZ4 and Fast LZ perform best in speed and resource efficiency, especially with RAM caching. At the same time, 7-Zip/LZMA achieves the highest compression ratios at the cost of slower loading. Brotli emerged as a strong option for web-based AR/VR content, striking a balance between compression efficiency and decompression speed, outperforming Gzip in WebGL contexts. The Addressable Asset system with LZ4 offers the most efficient balance for real-time AR applications. This study will deliver practical guidance on optimal compression method selection to improve user experience and scalability for AR digital twin implementations.

**Funding:** This work was supported by the Big Data Analytics Center of United Arab Emirates University under grant code G00004526.

**Competing interests:** The authors have declared that no competing interests exist.

# 1 Introduction

Industry 4.0 technologies, including Digital Twins, Artificial Intelligence, Virtual Reality, and Augmented Reality, are reshaping the industrial and education sectors [1–11]. These advancements offer intelligent, inclusive, and personalized learning experiences that align with global education objectives. By leveraging these technologies, education can be transformed into a more interactive and accessible realm for all.

Digital Twin (DT) technology has gained significant interest among companies and researchers due to its numerous benefits in industries and education. Although still in development, DT has garnered attention from various sectors, leading to multiple definitions introduced by researchers and industrial companies based on its applications and fields [12]. In 2023, Grieves classified digital twins into three subcategories: Digital Twin Prototype (DTP), Digital Twin Instance (DTI), and Digital Twin Aggregate (DTA) [13]. DT refers to a digital replication of a physical object connected to its virtual counterpart, such as a machine or its components. This connection allows data transfer between the physical and digital replicas, resulting in a responsive digital twin interacting with external factors like its physical counterpart. Meanwhile, DTP is a virtual representation or simulation of a physical object, process, or system. It goes beyond mere 3D modeling by incorporating real-time data and interactivity, creating a virtual dynamic digital counterpart of a non-existent physical entity. The use of digital twin technology has extended beyond the automotive industry and is now widely applied in various industrial sectors and urban services. For instance, manufacturing industries use digital twins for monitoring, development, operational forecasting, and virtual commissioning [14]. Furthermore, digital twins are employed to create virtual replicas of smart cities and for use in construction [15], advancing healthcare [16], optimizing oil and gas operations [17], enhancing automobiles [18], and even transforming education [19]. To further enhance the capabilities of digital twins, Extended Realities (XR) technologies such as Augmented Reality (AR) and Virtual Reality (VR) serve as complementary tools. VR enables users to experience simulated computer-generated environments resembling the physical world, while AR overlays digital images or 3D content in the real world through wearable or mobile devices. These XR technologies extend the potential of digital twins by digitally modeling physical objects and facilitating interaction with digital content. However, implementing AR/VR technologies, including digital twins, requires efficient devices equipped with high-performance hardware to maximize user experiences when displaying 3D models on users' devices. Herein lies a challenge, as mobile devices, though portable, often have limited computational power compared to computers. Moreover, memory limitations and storage issues can hinder the development of AR/VR-based applications on mobile devices.

Unity is an exceptional technology framework in game development, playing a pivotal role in creating 50% of worldwide games [20], and most studies have chosen it for AR/VR experiences [21]. The most common method of managing assets in Unity, the Resources Folder method, has limitations related to application size and management. As assets are fully packed into the application structure, the size of installation files increases, posing challenges due to capacity limitations on known publishing platforms like Google Play and Apple Store. Furthermore, frequent asset replacement and updates may inconvenience users, requiring them to re-download the complete app instead of targeted updates. To address these challenges, Unity has introduced two methods: the Assetbundle method and the recent Addressable Asset system [22]. The Assetbundle method compensates for the limitations of the Resources Folder method by including assets in a bundle called "AssetBundle". AssetBundles contain various resources, such as textures, materials, sounds, animation resources, text assets, scenes, and more. During the execution phase, the game requests the remote server to load the AssetBundle and utilize the resources

within. This method enables level updates, resource pack downloads, app updates, and efficient app size reduction. However, it has drawbacks concerning Bundle Dependency Management and updating the Asset Path in the bundle. On the other hand, Unity's Addressable Asset system is built upon the AssetBundle system to simplify the build and integration process [23]. It offers robust and simplified asset loading, leveraging the advantages of the Assetbundle method while overcoming its limitations. The Addressable Asset system utilizes addresses associated with assets to determine when to load and unload them, enhancing performance through reduced iteration time, improved memory management, and decreased initial scene load time. Unity's Addressable Asset system is a powerful tool built on top of the AssetBundle system, designed to streamline the build and integration process. It simplifies asset loading and offers several advantages over the traditional AssetBundle method. By utilizing addresses associated with assets, the Addressable Asset system optimizes performance by reducing iteration time, improving memory management, and decreasing the initial scene load time [23].

To achieve optimal performance in AR/VR-based applications, optimizing CPU time and RAM usage while effectively managing assets using the best available compression algorithms is essential. Unity provides three compression methods for AssetBundle packaging: LZMA, LZ4, and No Compression [24]. Among these, LZMA is a popular compression format that minimizes file size but has slower decompression, leading to longer loading times. On the other hand, Unity supports LZ4 compression, which achieves larger compression ratios while ensuring faster decompression [25]. However, on WebGL builds, LZMA AssetBundle compression is not supported and available for AssetBundles as WebGL does not support threading [26]. On the other hand, LZ4 is less commonly used in web-based AR experiences compared to formats like Gzip and Brotli. While LZ4 is efficient in terms of speed, it might not provide the same compression ratios as Gzip or Brotli., AR experiences in WebGL typically use compressed formats like Gzip and Brotli to optimize loading times and improve performance. These compression formats reduce the size of files transferred over the internet, resulting in faster loading times for AR content in web browsers. Brotli is a newer compression algorithm developed by Google that often provides better compression ratios than Gzip [27, 28]. Both formats are widely supported by modern web browsers and servers, making them suitable choices for delivering compressed AR content over the web. The utilization of data compression techniques in scientific computing has been a subject of interest for improving file I/O performance and optimizing Internet bandwidth for different purposes [29–39].

The structure of this article is delineated as follows: Section 2 provides an exposition of the foundational components derived from prior research that underpin the development of this study. Section 3 expounds upon the experiment's design and setup, elucidating the chosen compression methodology and the datasets employed in the experimental framework. Section 4 is dedicated to presenting and discussing the outcomes and their implications. The article culminates in Section 5, where a comprehensive conclusion is drawn.

## 2 Literature review

In an AR/VR-based environment, researchers mainly use AssetBundle with different compression methods to achieve the best performance, while the newly integrated Addressable asset management method is still not considered a better solution for the same purpose. Santos [40] suggests using AssetBundles, created with LZMA compression and LZ4 caching, to load 3D models, textures, and audio files at runtime, avoiding application compilation. This approach leads to advantages like lower memory consumption, faster downloads, and reduced storage space in the cache. The paper of Glushakov et al. [41] focuses on personalized AR experiences achieved through edge-based on-demand holographic content provision tailored to user

interests and conditions. They used 3D models in .obj format with different triangle counts from the Stanford 3D Scanning Repository to represent different model qualities in AR gaming along with PNG textures. The research used Unity's AssetBundle feature with LZMA compression to export 3D models and store them in a remote server.

On the other hand, Solmaz & Van Gerven [42] evaluated asset storage options in unity addressable asset manager: uncompressed, LZ4, and LZMA methods for developing computational fluid dynamics simulation digital applications to teach mixing in chemical engineering. The researcher concluded that LZ4 reduced bundle size by 1.58x and LZMA by 3.05x compared to uncompressed. Processing time had no significant difference between uncompressed and LZ4, but LZMA showed some delay during decompression. In their attempt to develop 3DWeb Applications for Participatory Urban Planning, Alatalo et al. [43] state that LZ4 is the default compression for WebGL bundles in Unity 5.5+. At the same time, LZMA is recommended for native builds. Browsers use Gzip compression with LZ4 and some support Brotli. However, the automated publishing of the bundles used in their study relies on Google Cloud Storage, which does not support Brotli, hindering performance assessment. Motivated by the implementation of different compression methods to aiming the best performance in storing and visualizing AR contents, this research paper is considered the first of its art in providing statical quantitative data as a part of an experimental DTP educational laboratory, aiming to explore this area by investigating and comparing the performance of AR-based applications on android mobile devices. Specifically, it focuses on evaluating different Asset bundles and Addressable methods for downloading and decompressing bundles using various lossless compression algorithms that are not supported by the standard Unity Assetbundle packages. This limitation restricts developers' flexibility in selecting the most suitable compression algorithm for their specific requirements. Furthermore, Unity provides a range of techniques and functions for handling asset bundles remotely, each tailored to different purposes. This study examines and analyzes some of these functions to gain insights into the behavior of AR-based applications in terms of RAM usage, CPU time, and file size. Additionally, mathematical models will be developed to aid in predicting the RAM and CPU time required to download and display prefabs from remote servers and estimating the compressed bundle size relative to its content, such as video files and 3D model polygon counts.

This study focuses on achieving the following goals:

- Researching and experimenting with various data compression techniques and algorithms to explore their effectiveness in reducing file sizes while maintaining data fidelity and thus enabling the correct tradeoff technique and conclusion.

- Identifying the optimal data compression algorithm suited for the specific data sets' specific characteristics, considering factors such as compression ratio, computational efficiency, and CPU time.

- Integrating different compression algorithms for bundle compression within the Unity environment, enabling seamless utilization of the chosen compression techniques while rendering the 3D model data.

- Modeling the relationship between the prefabs content and the compressed file size to facilitate bundle output size estimation and complete CPU time.

## 3 Materials and methods

A *bundle* is a popular file format used in game development to package and store various game assets, such as textures, models, sounds, and other resources. Due to these assets' large size and

complexity, efficient storage and transmission are crucial for optimal game performance. This is where data compression algorithms play a vital role. Data compression algorithms are techniques that reduce the size of data files without sacrificing essential information. They achieve this by removing redundant or irrelevant data, exploiting statistical patterns, and encoding the data more compactly. In asset bundles, compression algorithms are employed to decrease the overall file size, resulting in faster loading times, reduced memory usage, and improved performance.

Various compression algorithms are utilized for asset bundles, each with advantages and characteristics. Some commonly used lossless algorithms include:

- LZMA is a robust compression algorithm known for high compression ratios. It combines LZ77 and Markov chain methods and is widely used in applications like 7z archives.

- LZ4. This library uses an algorithm based on LZ77. It is a library focused on fast compression and decompression. It has 12 different compression levels.

- Fast LZ is a speedy, straightforward compression algorithm that prioritizes fast processing over maximum compression efficiency. It implements the LZ77 algorithm for lossless data compression that utilizes a sliding window approach and is commonly used in real-time data transmission and mobile platforms.

- 7-Zip provides a higher compression ratio than other file formats, using LZMA and LZMA2 compression algorithms. LZMA: LZMA is the default method for 7z compression.

- Brotli is a generic-purpose lossless compression algorithm that compresses data using a combination of a modern variant of the LZ77 algorithm, Huffman coding, and second-order context modeling, with a compression ratio comparable to the best currently available general-purpose compression methods.

- Gzip utilizes the DEFLATE compression algorithm, which combines LZ77 (Lempel-Ziv 77) and Huffman coding techniques to achieve efficient compression ratios. It strikes a good balance between compression ratio and decompression speed.

- Zip compression works by reducing the size of files using the DEFLATE algorithm; it provides a reasonable compression ratio while offering efficient decompression speeds.

The choice of compression algorithm for asset bundles depends on various factors, including the type of assets being compressed, target platform constraints, desired compression ratio, decompression time considerations, and trade-offs between file size reduction and visual or audio quality preservation. Game developers carefully evaluate these factors to select the most appropriate compression algorithms and settings for their asset bundles, aiming to balance efficient storage and transmission of assets and the quality and performance requirements of their games [44].

## 3.1 Compression methods

For testing purposes, various compressed assets are stored on a local server. These assets will be accessed by end users using Android devices through an AR-based application developed with Unity and the Vuforia AR-Engine. The application utilizes relayed target images to initiate the download process. Once the desired asset is downloaded, it undergoes decompression using Unity Asset-bundle, Unity Addressable, C# based native compression libraries, and an open-source compression plugin before being displayed on the users' screens.

The testing process involved two main stages that were outside the study's focus. Firstly, the entire data set was compressed to achieve the highest possible compression ratio. Secondly,

different parameters within the algorithms were modified and tested to optimize their performance and obtain the best results. Numerous algorithms and methods were investigated, considering their compression ratios and speeds for the given data sets; seven were chosen for testing purposes. These algorithms include 7-ZIP, Fast LZ, LZ4, LZMA, ZIP, GZIP, and Brotli. To evaluate these algorithms, each one was run on datasets of different sizes, ranging from 12 MB to 590 MB, and content. The compression ratio, mobile memory usage, and the time needed for asset visualization were documented for each test. Analyzing the results of these tests was instrumental in identifying the most appropriate algorithm or technique for the specific datasets, yielding valuable insights for the study.

## 3.2 System design

The testing benchmark consisted of several components, including the following:

- Testbeds: The test beds comprise an Android mobile phone, a computer with Unity installed, and a mobile application specifically designed for testing purposes.

- Data-sets: The data-sets used in the testing process were bundles containing 3D models and video media. These bundles served as the input for the compression techniques being evaluated.

- Compression techniques: The compression techniques employed in the testing system encompassed Mono Behaviour C# scripts and Android Libraries responsible for compressing and decompressing the data sets on the testbeds.

Mono Behaviour C# scripts and libraries containing different compression algorithms and loading functions were installed within the benchmark app installed on the testbeds. When the compressed bundled data sets were loaded, the testing environment was initiated, and the selected algorithms were executed on the files. The CPU time required and the maximum RAM usage during the test were recorded, serving as the results for this study.

## 3.3 Testbed

The testbeds utilized in this study consisted of a central computer, an Android mobile device, and Unity software applications and tools. These testbeds were crucial for running and evaluating the performance of various compression techniques on the data sets. The primary testbed was a Windows-based computer running the Unity software, which served as the development environment for the principal benchmark application based on augmented reality (AR). Additionally, the computer hosted the open-source, cross-platform Apache web server solution stack package v3.3.0 (XAMPP) [45], responsible for hosting and transmitting the data upon user application requests.

The test benchmark application was developed using C# programming language, utilizing Unity v2021.3.12f1 and Vuforia AR-Engine v10.3.2. Vuforia AR-Engine [46] facilitated target image recognition within the application. The final version of the test benchmark application was configured and built on a Lenovo mobile phone running Android 8.0 with API level 26. The application was built using an IL2CPP scripting backend and targeted the ARM64 architecture. Hardware specifications for the testbed can be found in Table 1.

## 3.4 Data-sets

A comprehensive analysis was conducted using a total of 12 data sets, which were imported into the Unity environment for evaluation. These data sets consisted of three distinct industrial virtual laboratory 3D models with varying polygon counts. Polygons are the building blocks

**Table 1. Hardware technical specifications of utilized testbeds.**

| Device | Lenovo K5 play | Main Computer |
|---|---|---|
| **Model** | Lenovo L38011 | DESKTOP |
| **CPU** | Qualcomm Snapdragon 430 1.4GHz Octa-Core | Intel(R) Core(TM) i7-7700K CPU 4.20GHz 4.20 GHz |
| **RAM** | 3 GB | 48.0 GB, 2400 MHz |
| **ROM** | 32GB | 500 GB SSD |
| **System** | Android v8.0.0 | Windows 10 Pro 64-bit |
| **Camera** | 13.0MP | Logi C615 HD WebCam |
| **Display** | 5.7 inch | 27" FHD Monitor |
| **Resolution** | 1440 x 720 | 1920 x 1080 |
| **WLAN/ NIC** | WiFi, 802.11 a/b/g/n Rate: 450 Mbps | Ethernet, Realtek PCIe GbE Family Controller, Data rate: 2.5GB/sec |

that define the surfaces of 3D models, while vertices represent the points where polygons meet. The connectivity between vertices and the order in which they are connected form the geometry and topology of the 3D model. Polygons and vertices define the shape, structure, and appearance of 3D objects in computer graphics.

The study's objective was to investigate the performance of compression algorithms when dealing with diverse data sets. Therefore, modifications were made to the 3D model structure and shape, resulting in different vertices counts. The data sets were also populated with various MP4 (3840x2160) media content of different sizes to simulate an authentic educational environment where educational videos are utilized within an augmented reality (AR) setting. Other data types, such as font, texture, and Mono Behaviour scripts, were ignored due to their small size.

Table 2 provides detailed information about each asset, including its size and content type, showcasing the diverse nature of the data sets used in the study.

The combination of altered polygons, different vertices, and varied media content enabled a comprehensive examination of the compression algorithms' behavior and effectiveness when dealing with various educational assets in an AR environment.

**Table 2. Main dataset content.**

| Targets | Bundle Size [Mbyte] | 3D Model Polygons Count | 3D Model Vertices Count | Video Size [Mbyte] |
|---|---|---|---|---|
| T1 | 12.40 | 204679 | 309991 | 0 |
| T2 | 49.90 | 586339 | 1229170 | 0 |
| T3 | 105.00 | 586339 | 1229170 | 40.1 |
| T4 | 166.00 | 586339 | 1229170 | 98.9 |
| T5 | 256.00 | 586339 | 1229170 | 188 |
| T6 | 383.00 | 586339 | 1229170 | 315 |
| T7 | 492.00 | 586339 | 1229170 | 425 |
| T8 | 188.00 | 2756466 | 4720489 | 0.0 |
| T9 | 293.00 | 2756466 | 4720489 | 43.6 |
| T10 | 436.00 | 5513798 | 9442203 | 6 |
| T11 | 475.00 | 5513798 | 9442203 | 44 |
| T12 | 589.00 | 8340062 | 12193656 | 44 |

### 3.5 Compression techniques

In this study, 12 datasets were packed into .*bundle* format using the Unity packages "Asset-Bundle Browser Tool" and "Addressable Package" version 1.19.19. These datasets were compressed into LZ4, LZMA, and Uncompressed bundles. The uncompressed bundles were further compressed using a native C# script that utilized open-source libraries and a third-party plugin available in the Unity Asset Store. The resulting compressed bundles were stored on a local Apache server running on the main computer, as shown in Fig 1.

PolyCount v1.04 [47] unity editor extension was employed to calculate the number of vertices and polygons in the 3D models. For achieving 7z/LZMA2 compression, the open-source file archiver 7-Zip version 23.01 for Windows 64x was employed. The compression engine parameters were carefully chosen to obtain the best compression results, including settings for Fast LZ, LZ4, LZMA, ZIP, Gzip, and Brotli formats.

### 3.6 Mono behaviour of loading methods

In Unity, Mono Behaviour is a class that serves as the base class for scripts that control the behavior of the same Objects within a Unity scene. Mono Behaviour provides a range of methods and functionalities that can be utilized for various purposes, including loading assets.

The downloaded asset bundles can be cached in memory or saved on a local disk before decompression. Caching asset bundles in memory results in faster loading times but consumes a significant amount of memory, while using a disk cache helps preserve memory at the cost of increased loading times [25]. Both approaches were tested in this study. Unity provides different functions for downloading and decompressing asset bundles. To ensure consistency in evaluating the decompression speed and memory consumption of the compression algorithms

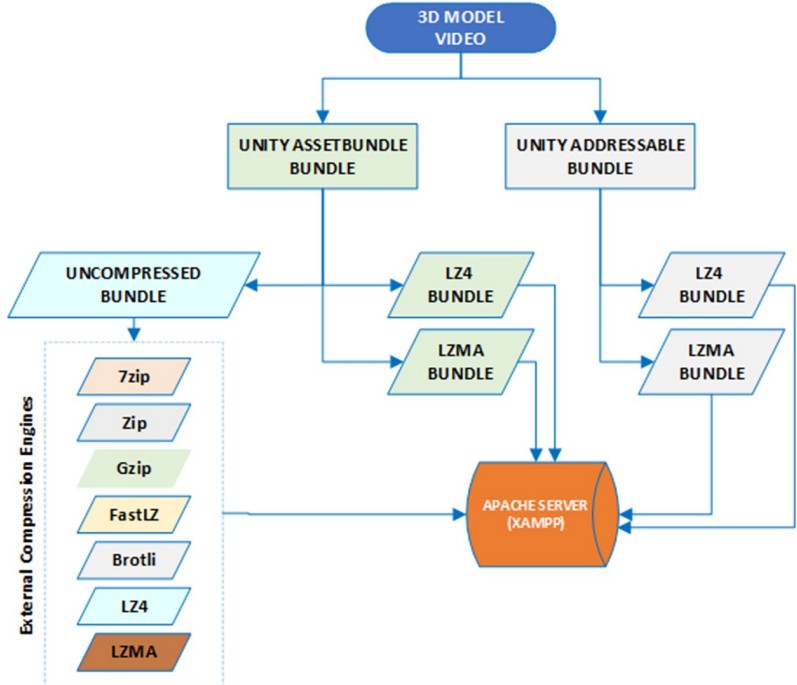

**Fig 1. Asset bundle compression model.**

used in the testbed, specific functions were chosen that separate the downloading process from the decompression process to measure performance metrics accurately.

When the end user's mobile device recognizes a pre-configured target image, a download Coroutine is initiated. After displaying the prefab, the bundle is unloaded, or the prefab game object is destroyed once the target image is lost and reloaded when the target image is found again, enabling the study of memory behavior with different approaches. Various functions and classes were employed to handle bundles and prefabs' download, loading, and unloading.

- AssetBundle_Disk: To cache bundles on the local disk, the WebClient class for downloading bundles was used for Unity AssetBundle and external compression algorithms on disk caching methods. WebClient was chosen over UnityWebRequest due to observed memory leaks while downloading assets using UnityWebRequest. The downloaded AssetBundle is temporarily stored in the application's *persistentDataPath* for further processing. Once the download is complete, *AssetBundle.LoadFrom-FileAsync()* is called to load the asset, followed by *AssetBundle.LoadAssetAsync<GameObject>* to decompress the AssetBundle. The 3D model with its associated objects is instantiated and displayed on the end user's mobile device. Unneeded assets and handlers are released to clean up the memory. *GC.Collect()* enforces garbage collection, while *AssetBundle.UnloadAllAsset-Bundles()* and *Resources.-UnloadUnusedAssets()* are employed to free memory from loaded bundles and unused assets. The cloned prefab game object is destroyed when the target image is lost using *DestroyImmediate()*.

- External Compression Engine: Similar to the AssetBundle_Disk version, the same functions and procedures are followed, requiring an additional step to decompress the uncompressed bundle from the archive. The downloaded archived bundle is first decompressed using the related decompression function before loading the uncompressed AssetBundle using *Asset-Bundle.LoadFrom-FileAsync()*. To avoid high CPU frame rates caused by calling the decompression function on the main thread, it is configured to run on a parallel thread, and a handler is used to report the status to the main thread for starting the loading and instantiation of the prefab, as Unity AssetBundle loading procedures must be called from the main thread. The unloading and destruction of bundles/prefabs follow the same approach as AssetBundle_Disk.

- Addressable version: The *Addressables.Download-DependenciesAsync()* method with Addressable label reference is used for downloading assets. Upon completing the download process, this operation is released, and the AssetBundles are stored in the engine's AssetBundle cache. *Addressables.LoadAssetAsync<GameObject>* is used to load and decompress the AssetBundle into memory, while *Instantiate()* is employed to clone and display the uncompressed bundle. The loaded asset is released using *Addressables.Release()*, while the cloned prefab is destroyed immediately after the target image is lost.

- AssetBundle_RAM: Both, *UnityWebRequestAssetBundle.-GetAssetBundle()* and *Download-HandlerAssetBundle.Get-Content()* without using a hash or version identifier is configured to download and cache the AssetBundle, which is then extracted and decoded once enough data has been downloaded. As the uncompressed bundle is cached, the AssetBundle is loaded, and the prefab is directly instantiated using *AssetBundle.LoadAsset()* and *Instantiate ()*. Compared to the previous approaches, the uncompressed bundle remains loaded in memory, but the cloned prefab is destroyed. This results in faster redisplaying of the prefab on the mobile device.

**Table 3. Summary of mono behaviour of loading methods.**

| Loading Method | Compression | Abbreviation | Key Features |
|---|---|---|---|
| Unity AssetBundle_ Disk | LZ4<br>LZMA | ASS_D_ LZ4<br>ASS_D_ LZMA | Caches AssetBundles on local disk,<br>WebClient class for downloading and AssetBundle loading,<br>Implements decompression and memory management |
| Unity AssetBundle_RAM | LZ4<br>LZMA | ASS_R_LZ4<br>ASS_R_LZMA | Caches AssetBundles in memory,<br>UnityWebRequestAssetBundle.GetAssetBundle()<br>DownloadHandlerAssetBundle.GetContent() |
| Unity Addressable | LZ4<br>LZMA | ADD_LZ4<br>ADD_LZMA | Addressables.DownloadDependenciesAsync(),<br>Stores AssetBundles in the engine's AssetBundle cache |
| External Engines | 7 Zip<br>Brotli<br>Gzip<br>Zip<br>Fast LZ<br>LZMA<br>LZ4<br>.bundle | 7 Zip<br>Brotli<br>Gzip<br>Zip<br>Fast LZ<br>LZMA<br>LZ4<br>Un | Caches Archived AssetBundles on local disk,<br>The archived bundle decompressed using external engines,<br>Asset bundle used to encapsulate and display prefab |

To avoid cached versions on mobile devices, the caching memory is freed, and all previously downloaded versions are deleted when the benchmark app starts using *Caching.ClearCache()* and *Addressables.ClearDependency-CacheAsync().* Table 3 summarizes the methods implemented along with their features.

## 3.7 Performance evaluation metrics

The decompression and prefab loading process involves loading the respective file into memory. The efficiency of this process is contingent upon the available RAM and the size of the file being decompressed. Consequently, the performance metrics chosen provide insights into the impact of these factors on the overall system performance. The Compression Ratio (CR) is a critical performance metric concerning the required download time. CR quantifies the effectiveness of a compression algorithm in reducing the size of the compressed data when compared to the original uncompressed data. Eq 1 is utilized to calculate CR, allowing for the determination of the achieved reduction.

$$CR = \frac{Uncompressed\ Bundle\ Size}{Compressed\ Bundle\ Size} \tag{1}$$

In the context of memory usage during bundle decompressing and prefab loading, various memory parameters are measured to assess memory consumption. Like the compression ratio, memory usage is influenced by the characteristics of the employed algorithm. The following memory parameters are typically considered: Total Used Memory, System Used Memory, and Total Reserved Memory.

The Total Used Memory refers to the amount of Memory utilized and monitored by the Unity engine. It represents memory consumption directly attributed to the execution of the bundle decompression and prefab loading operations. System Used Memory indicates the Memory reported by the operating system (OS) as being actively used by the application. This value encompasses the memory resources allocated and utilized by the app beyond the scope of the Unity engine.

Total Reserved Memory represents the Memory allocated by Unity for tracking purposes and pool allocations. It includes Memory dedicated to managing internal processes and maintaining efficient memory utilization. This study's key focus is the complete time required to

display the target prefab. Given that the variation in compression ratio is directly related to the bundle's content, the time required for downloading the compressed bundle and subsequent decompression will differ accordingly. The total time ($T_t$) can be theoretically calculated using Eq 2:

$$T_t = T_d + T_{un} + T_{de} + T_{in} \tag{2}$$

Where: $T_t$ represents the overall time taken to display the prefab on a mobile device, $T_d$ denotes the time required for downloading the compressed/achieved asset bundle, $T_{un}$ signifies the time taken for decompressing the achieved asset bundle, $T_{de}$ refers to the time taken for loading or decompressing the asset bundle, $T_{in}$ represents the time taken for instantiating and displaying the prefab.

In the context of External $E$ compression algorithms, $T_{un}$ (time taken to decompress the achieved asset bundle) is applicable. This is because the uncompressed asset bundle is first archived into a specific file format corresponding to the compression method used (i.e., 7z, zip, br, etc.). When utilizing Assetbundle and Addressable methods, the asset bundle is decompressed directly upon calling the LoadAsset() method. By employing this formula, the comprehensive time required for the entire process of displaying the target prefab can be evaluated. This includes the time spent downloading, decompressing, loading, and finally rendering the prefab on the mobile device.

## 3.8 Data sample collection

To mitigate the impact of uncontrollable factors on the obtained results, 840 measurements were conducted using the custom benchmark application. As the download speed of compressed asset bundles can vary depending on the hardware performance of both the mobile device and the server, each data set of the 12 asset bundles was tested five times at different time intervals for each compression method. The average of these measurements was considered for specific performance metrics. The testing process involved three phases:

1. Downloading Phase: The benchmark application initiated the download process for each asset bundle using the respective compression method. The time taken to download the asset bundle was recorded using *Stopwatch()* function for performance evaluation.

2. Decompression Phase: The decompression process took place once the asset bundle was successfully downloaded. The benchmark application employed the appropriate decompression algorithm for each compression method to extract the contents of the asset bundle. The time required for decompression was measured and analyzed as a performance metric.

3. Loading Phase: The benchmark application loaded the assets from the decompressed bundle after decompression. This phase involved loading the required textures, models, audio clips, or other media elements within the asset bundle. The loading time was recorded to assess the performance of each compression method.

By conducting multiple measurements and considering the mean results, the study aimed to minimize the impact of external factors and obtain reliable performance metrics for the different compression methods used.

To compare the performances of different algorithms, an initial set of testing was conducted using asset bundles that contained the same 3D model but with varying sizes of video files. This set of files, labeled 1-6, ranged from 12.4 MB to 466.5 MB. End users using their smartphones obtained measurements of time and RAM usage, and the data was stored in the

smartphone's internal memory by capturing screenshots. Additionally, the smartphone was connected to a central computer running Unity software via USB to record debugging and logging information from the benchmark application for further analysis. The data collected was then analyzed on the computer using the integrated Unity Profiler. R-Studio v2023.03.1 Build 446 [48] and Tableau Desktop v2019.3.3 [49] were utilized to establish correlations and perform statistical analysis. Various models, including Linear, Polynomial, and Exponential Regression, were employed to create two-dimensional graphics and determine the relationships between the different metrics used. By employing these analysis tools and statistical models, the study aimed to understand the correlation between metrics such as file size, decompression time, RAM usage, and overall performance. This analysis provided valuable insights into the performance characteristics of the different algorithms tested. Fig 2 illustrates the flowchart of data collection.

## 4 Results and discussion

### 4.1 Compression ratio

The aim is to understand how the compression ratio, which represents the reduction in file size achieved by compression, is affected by the input data size. The relationship between data size and compression ratio, where larger datasets tend to exhibit improved compression ratios, is a widely acknowledged concept in the field of data compression due to different factors, such as Redundancy and Patterns, Context modeling, and Statistical properties. As data size increases, statistical regularities and distributions become more pronounced. Compression algorithms can leverage these properties to encode the data more efficiently and achieve better compression ratios. However, the relationship between data size and compression ratio is not always linear or consistent across different data types, as illustrated in Fig 3. Certain types of data, such as text or highly structured files, tend to compress more efficiently with increasing data size due to their inherent patterns. On the other hand, already compressed files or highly random data may show diminishing returns in compression ratio improvement as the data size increases.

The results obtained from employing various data compression algorithms revealed that 7-Zip and LZMA algorithms consistently yielded the highest compression ratios, ranging from 7.7 to 1.1. In contrast, the Fast LZ and LZ4 algorithms produced lower but consistently observed compression ratios. To gain further insights into the relationship between the data structure and the achieved compression ratios, the data was organized and grouped into different sets for more comprehensive analysis and examination. The relationship between the Compression Ratio and the bundle content of lossless compression algorithms was investigated. Linear and nonlinear regression analyses were performed to analyze the data compression with different content.

**4.1.1 Relationship between compressed size and video size: A regression analysis.** For the dataset labeled "First Data Set" shown in Table 4, the relationship between the compressed file size and the content of the data type "MP4 video" was examined. Since Unity uses the same compression engine, the data set comparison will include the Addressable method running on different algorithms.

The results indicated that this relationship could be effectively modeled using a linear regression model, which was statistically validated. In Fig 4, an Asymptotic Regression Model showcased an exponential relationship between the file content (video and 3D model size) and the Compression Ratio (CR).

The constant term represents the compression ratio of the 3D model where the video size = 0. The findings indicated that all lossless compression methods had a minor

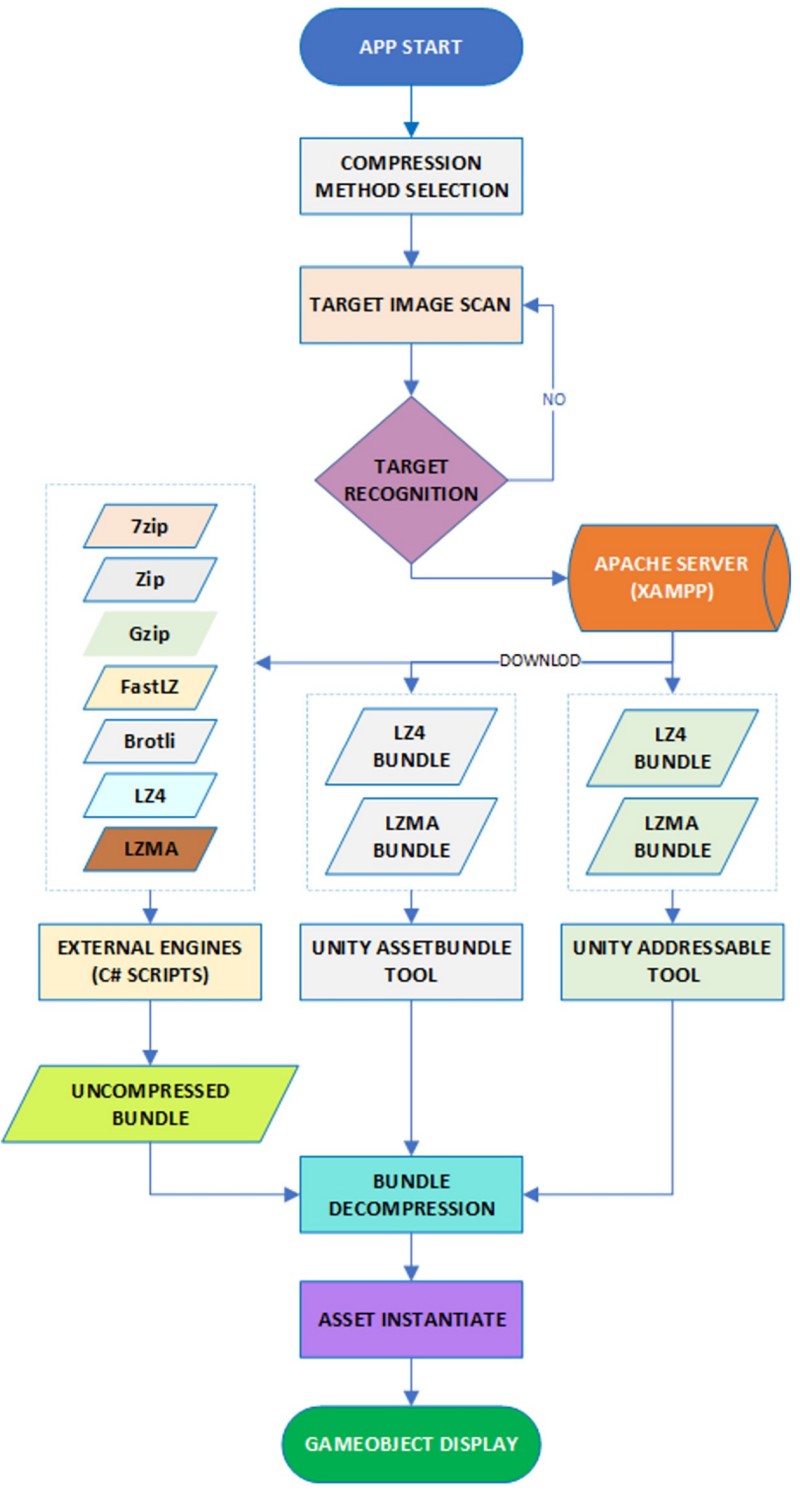

**Fig 2. Data collection flowchart.**

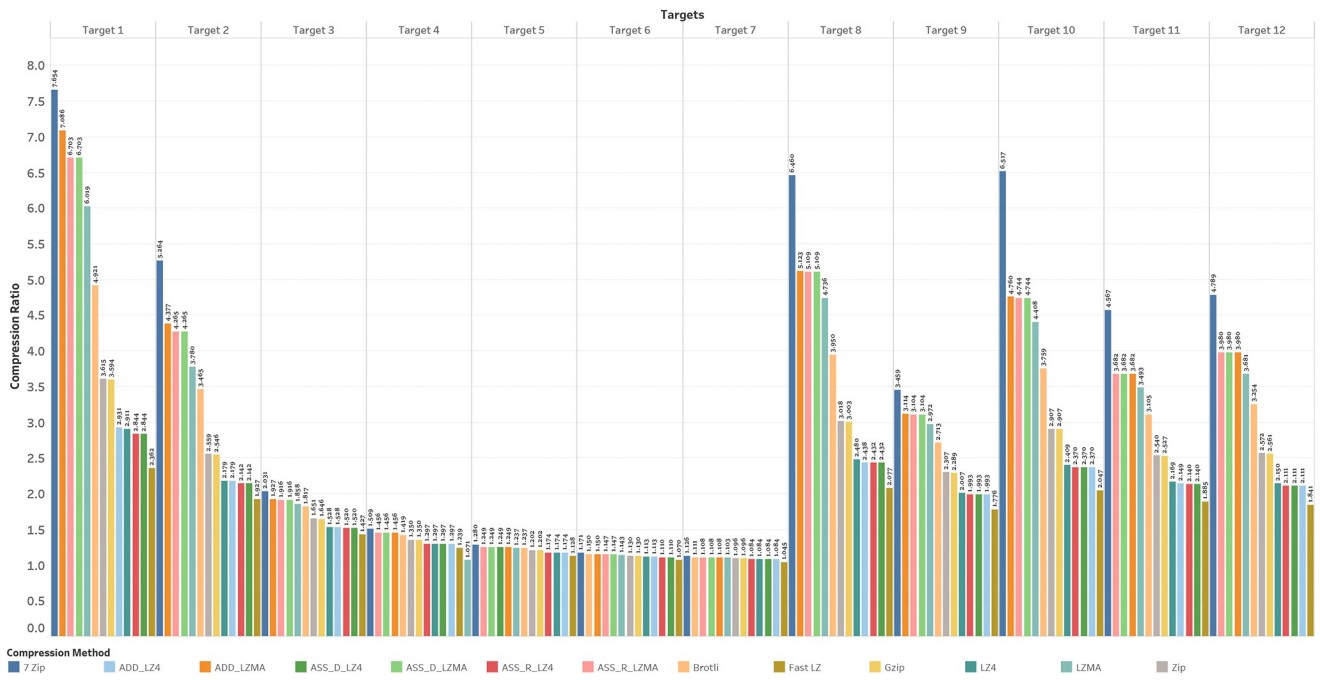

**Fig 3. Compression ratio.**

compression impact on files containing incremental MP4 video size while the CR varies on data containing a 3D model.

Fig 5 presents a plot demonstrating the relationship between the increase in video size and the corresponding linear increase in compressed file size.

The overall quality of the linear regression fit is assessed and validated using R-squared ($R^2$), Significance of Coefficients, and Residual Standard Error (RSE). In simple terms, the RSE is a metric used to evaluate how effectively a regression model fits a data set by quantifying the standard deviation of the residuals [50]. In evaluating various regression models, the residual standard error plays a valuable role, allowing for a meaningful comparison of their fits. It is calculated as illustrated in Eq 3:

$$RSE = \sqrt{\sum_{i=1}^{n} \frac{(y - \hat{y})^2}{df}} \tag{3}$$

where; $y$ is the observed value, $\hat{y}$ is the predicted value, and $df$ is the degree of freedom.

**Table 4. First data set.**

| Targets | Bundle Size [Mbyte] | 3D Model Polygons Count | 3D Model Vertices Count | Video Size [Mbyte] |
|---|---|---|---|---|
| T2 | 49.90 | 586339 | 1229170 | 0 |
| T3 | 105.00 | | | 40.1 |
| T4 | 166.00 | | | 98.9 |
| T5 | 256.00 | | | 188 |
| T6 | 383.00 | | | 315 |
| T7 | 492.00 | | | 425 |

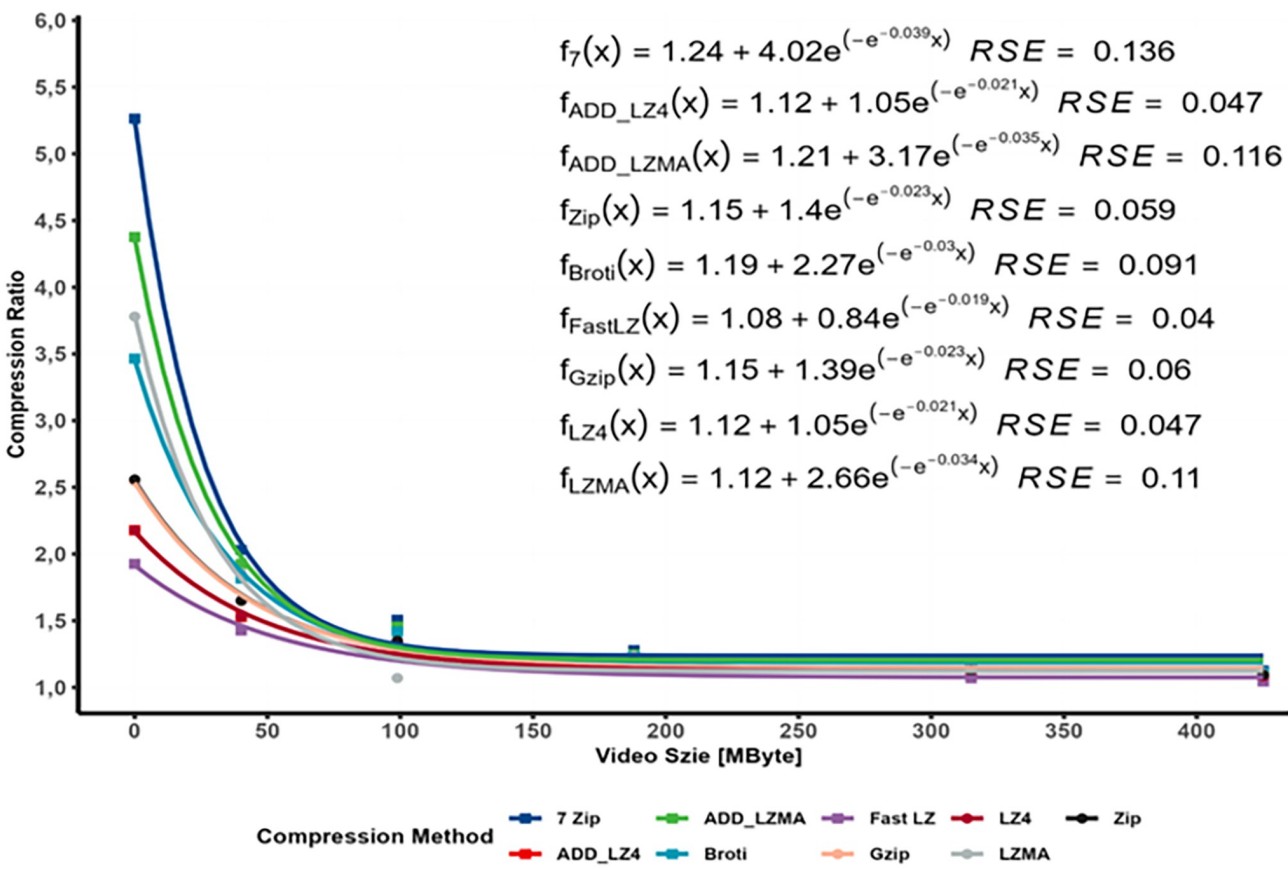

**Fig 4. Asymptotic regression model.**

The analysis revealed a strong linear correlation and significance level with a $R^2$ value of approximately 1 and a small *RSE*, consistent across all compression methods employed. In conclusion, this result highlights that lossless compression methods exhibit a slight compression impact on files containing MP4 video content. The relationship between compressed file size and video size demonstrates a linear trend with a coefficient of approximately 1, which reflects a very slight impact on the compressed size. At the same time, the intercept constants represent the CR of the 3D model size, which will be analyzed separately. On the other hand, the *CR* relationship with the file content can be modeled using an exponential model. These findings provide valuable insights into the effectiveness of various compression algorithms for MP4 video data and can be used to build a final model.

**4.1.2 Relationship between compressed size and 3D model characteristics.** In the second Data Set as shown in Table 5, the asset bundles are grouped and consist of 3D models with varying polygons and related vertices counts without video content.

To examine the relationship between the independent variable (3D model characteristics) and the compressed size as the dependent variable, a linear regression model is used to compare the relationship between the compressed bundle size and the vertices count as well as the polygons count, as depicted in Fig 6.

Both independent variables show a strong influence on the dependent variable, with high $R^2$ values indicating that these variables can explain a significant proportion of the variance in the compression size. However, the RSE varies among the models. For comparison purposes,

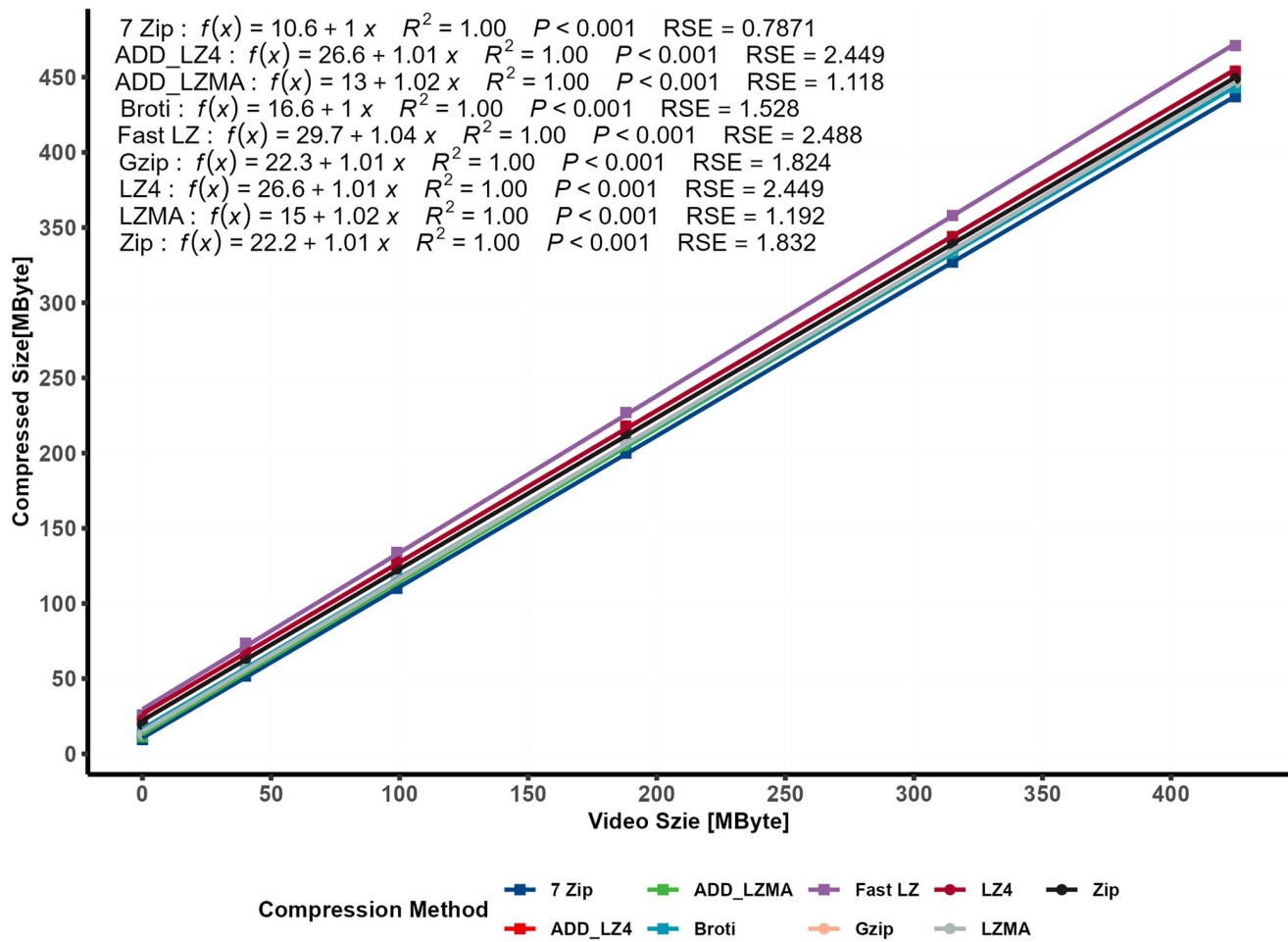

**Fig 5. Relationship between compressed size and video size.**

the RSE is also represented as a percentage by dividing the RSE for each compression method by the means of the actual y values (compression size) and multiplying by 100 as shown in Eq 4. The resulting percentage error will provide a measure of the average deviation of the model's predictions from the actual data relative to the mean of the data. A lower percentage error indicates a better fit of the regression model to the data set.

$$Error\ RSE\ (\%) = \frac{RSE}{mean(y_{actual})} \qquad (4)$$

**Table 5. Second data set.**

| Targets | Bundle Size [Mbyte] | 3D Model Polygons Count | 3D Model Vertices Count | Video Size [Mbyte] |
|---|---|---|---|---|
| T1 | 49.90 | 204679 | 309991 | 0 |
| T2 | 105.00 | 586339 | 1229170 | 0 |
| T8 | 166.00 | 2756466 | 4720489 | 0 |
| T10_2 | 418.00 | 5513798 | 9442203 | 0 |
| T12_2 | 545.00 | 8340062 | 12193656 | 0 |

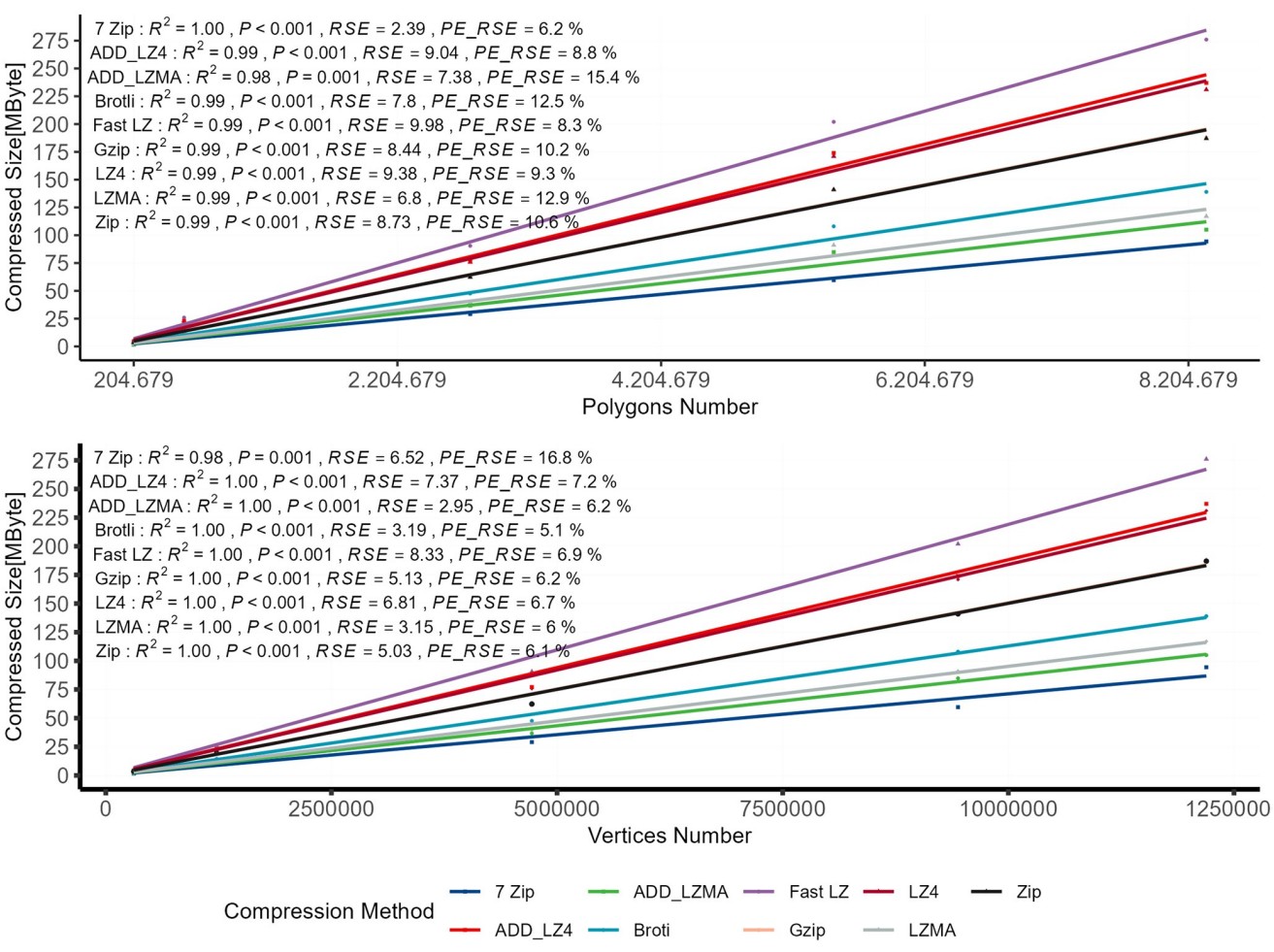

**Fig 6. Comparison of polygons and vertices count.**

The Percentage Error RSE (PE_RSE) reveals better results when using vertices count as a predictor compared to polygon count for all compression algorithms. However, interestingly, the 7 Zip algorithm exhibits a 10% lower RSE error percentage when using polygons as a predictor. Furthermore, AIC (Akaike Information Criterion) and BIC (Bayesian Information Criterion) are used as additional statistical measures for model selection in various statistical models, such as linear regression. They help balance goodness-of-fit and model complexity. AIC finds the best model by penalizing complexity less, while BIC applies a more decisive penalty for complex models. Lower AIC or BIC values for models with vertices count indicate better model fit. The RSE, along with AIC and BIC tests, can be utilized as quality indicators to confirm the hypothesis that the vertice count of 3D models can be used to determine the compressed file size for the selected compression algorithms. As a result, the vertices count will be considered for further analysis.

Table 6 includes all statistical terms and values required for building the model using the Eq 5:

$$y_i = \beta_1 \, x_{i1} + \epsilon \tag{5}$$

**Table 6. Summary of sample linear regression analysis conducted for compressed bundle size and vertex count.**

| | Vertices | Confidence Interval | $R^2$ | AIC | BIC | RSE |
|---|---|---|---|---|---|---|
| **LZMA** | 9.6e-06*** | [8.7e-06, 1.1e-05] | 1.0 | 29.1 | 27.9 | 3.15 |
| **LZ4** | 1.9e-06*** | [1.7e-05, 2.1e-05] | 1.0 | 36.8 | 35.7 | 6.81 |
| **Zip** | 1.5e-06*** | [1.4e-05, 1.7e-05] | 1.0 | 33.8 | 32.6 | 5.03 |
| **7-Zip** | 7.4e-06*** | [5.3e-06, 9.4e-06] | 0.98 | 36.4 | 35.2 | 6.25 |
| **Brotli** | 1.1e-05*** | [1.1e-05, 1.2e-05] | 1.0 | 29.2 | 28.1 | 3.19 |
| **Fast LZ** | 2.3e-05*** | [2.0e-05, 2.5e-05] | 1.0 | 38.8 | 37.7 | 8.33 |
| **Gzip** | 1.5e-05*** | [1.4e-05, 1.7e-05] | 1.0 | 34.0 | 32.8 | 5.13 |
| **ADD_LZMA** | 8.8e-06*** | [7.9e-06, 9.7e-06] | 1.0 | 28.5 | 27.3 | 2.95 |
| **ADD_LZ4** | 1.9e-05*** | [1.7e-05, 2.2e-05] | 1.0 | 37.6 | 36.4 | 7.37 |

+ p<0.1,

\* p<0.05,

\*\* p<0.01,

\*\*\* p<0.001

where; $i$ is the compression algorithm, $(y_i)$ number of dependent variables (compressed set), $\beta_i$ slope coefficient for explanatory variable, $x_{i_1}$ explanatory variable (Vertices count), and the $\epsilon$ is the model's residual error.

## 4.2 Total time

The cumulative time $T_t$ needed to showcase the target asset on the end user's device is illustrated in Fig 7, representing the conclusive outcome of the experimental measurements. The graph in Fig 7 encompasses all the parameters from Eq 2. The download time, which was consistent on average across all experiments, was validated using an incremental linear slope with an average download speed of 4 to 7Mbps. The variance of the download time of individual bundles concerning their sizes was too small (46 ms) and hence was ignored. The relationship between the uncompressed original bundle and the complete time appears to follow a linear regression pattern, albeit with varying slopes due to differences in compression ratios, download time, and decompression speed, which warrant further analysis.

Among the different compression methods, the fastest method to display the target asset is achieved using RAM caching with LZ4 compression (ASS_R_LZ4). In contrast, LZMA compression algorithms require significantly more time than other algorithms. In fact, LZMA is considerably slower than LZ4, with LZMA taking nearly double the time of LZ4 when using the Assetbundle method. The relationship between the original file size and the complete time is statistically significant for all datasets, except when LZMA compression is used. By examining individual models, the complete time can be predicted using linear models with the uncompressed bundle size as an independent predictor variable, as depicted in Fig 8.

When LZMA is used in different methods, the R-squared values of the linear models range between 0.67 and 0.87, with RSE ranging from 13 to 28 seconds. In contrast, utilizing the uncompressed bundle and ADD_LZ4 methods yields stable results with an RSE of only 5.7 and 4.7 seconds, respectively, representing a PE_RSE of 11%. However, the RSE percentage (PE_RSE) resulting from using LZMA can be considered extremely high, while all other compression methods exhibit acceptable error percentages ranging from 11% to 19%. For predicting the total time required for displaying the compressed bundles based on their content,

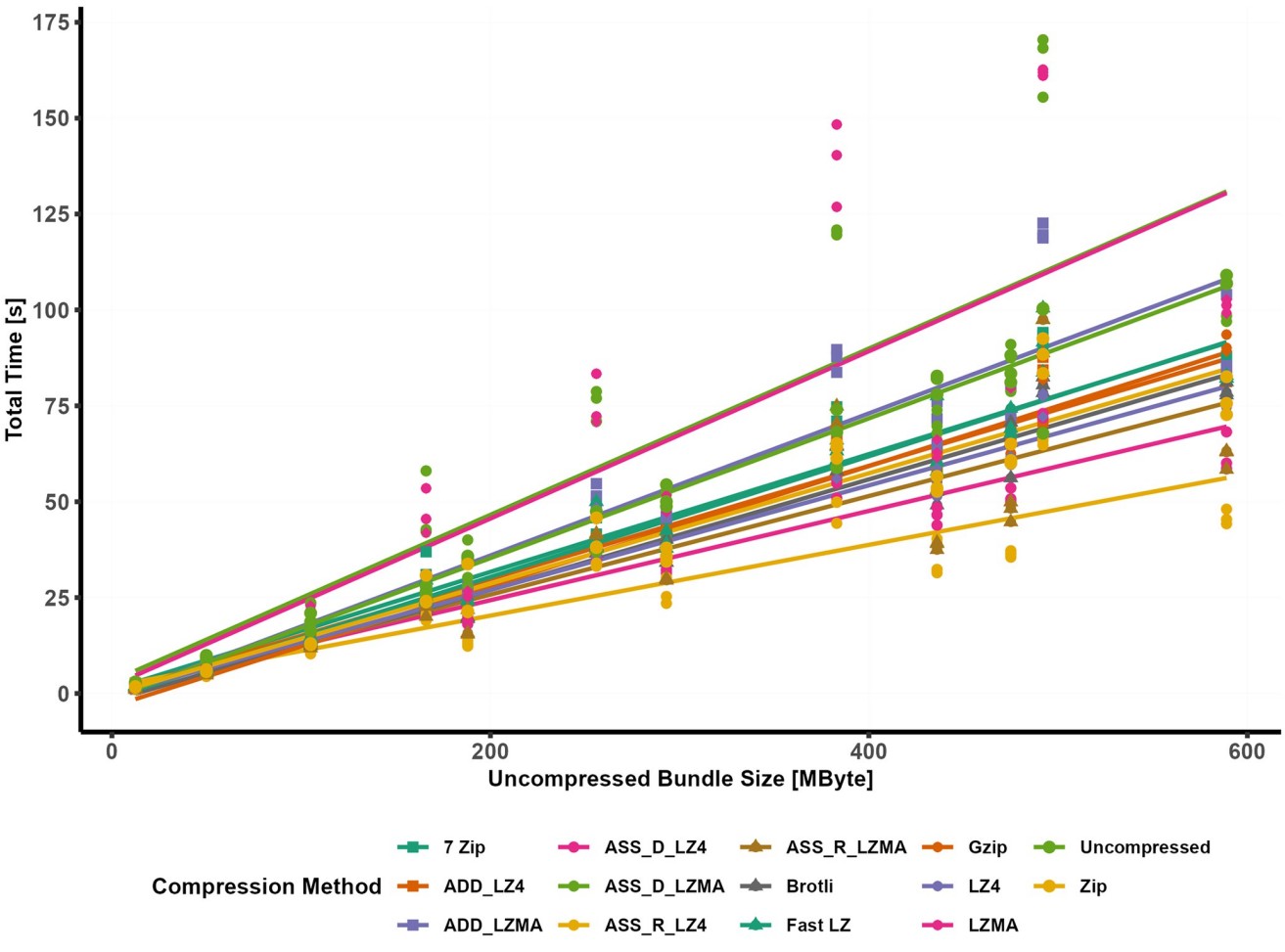

**Fig 7. Total time comparison.**

multiple linear regression is shown in Fig 9. In contrast, Fig 10 illustrates the differences in RSE values among different compression methods. It can be observed that the R-adjusted value and the RSE are improved in the regression between the total time as the dependent variable and the vertices number and the video size as independent predictors.

To generalize the model, the download time is removed from the model as the download speed might vary from one connection to another. This is done by subtracting the download time from the total time before modeling. However, the time required to download a file over the internet can be estimated using Eq 6 for download time, where Download Speed is the internet connection speed.

$$Download\ Time(s) = \frac{File\ Size\ (bytes)}{Download\ Speed\ (bytes\ per\ second)} \tag{6}$$

Fig 11 and Table 7 display the linear model along with the compression methods equation, which is created using Eq 7 to estimate the time required by each compression method to decompress and instantiate the Gameobject on the end user's mobile device.

$$y_i = \beta_1 x_{1i} + \beta_2 x_{2i} + \epsilon \tag{7}$$

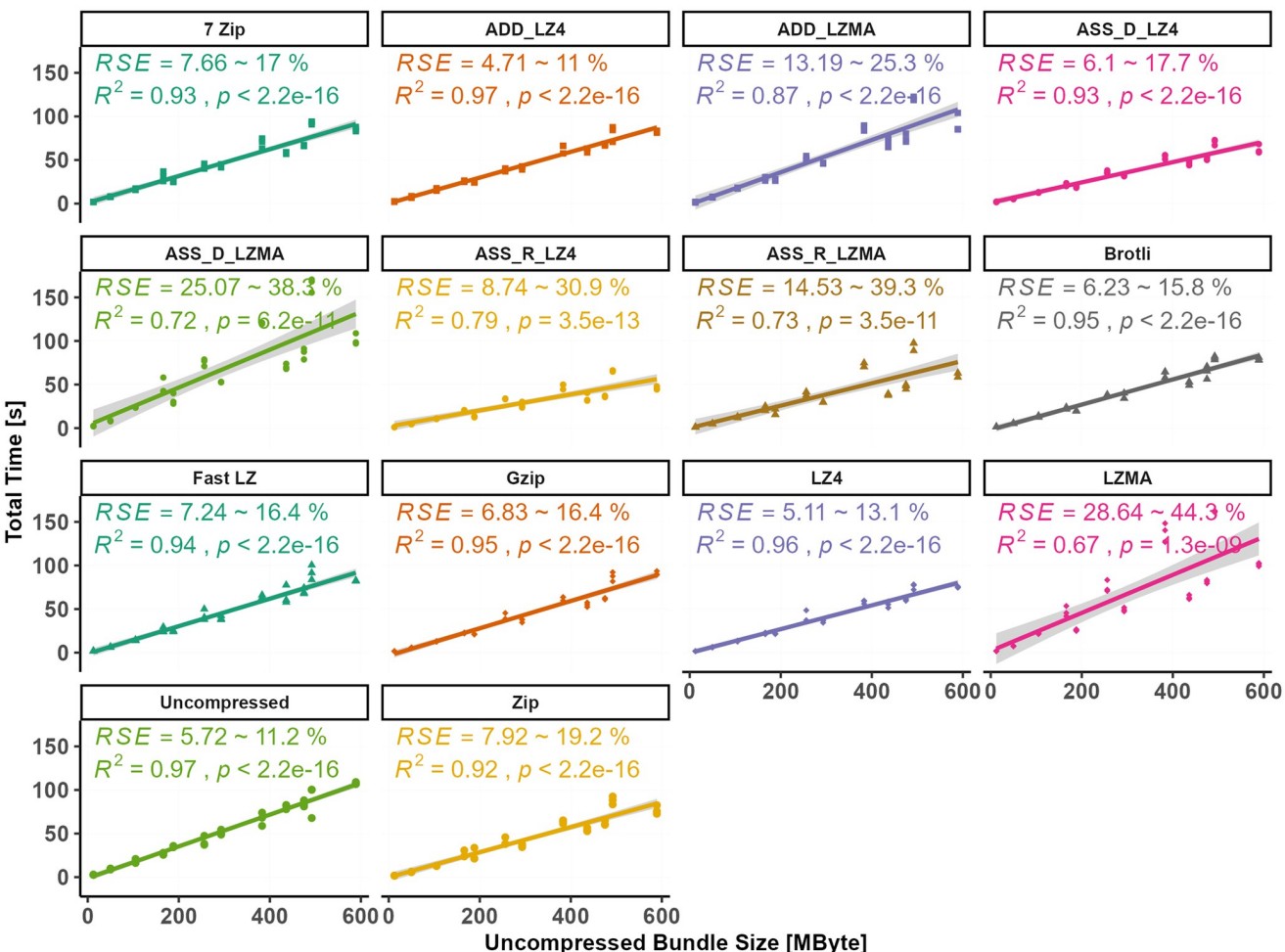

**Fig 8. Individual relationship between total time and bundle size (linear regression).**

where; $i$ is the compression algorithm, $(y_i)$ dependent variables (total time), $\beta_i$ slope coefficient, $x_1$ explanatory variable (Vertices count), $x_2$ explanatory variable (Video Size), and the $\epsilon$ is the model's residual error.

## 4.3 Decompression time of external compression engines

For evaluating the decompression speed of various nonsupported/integrated compression algorithms in Unity Assetbundle and Addressable tools, the decompression time required to decompress archived bundles used by the external engines is measured. Fig 12 illustrated a linear relationship between the non-compressed bundle size and decompression time, along with their respective linear models. LZ4, Fast LZ, ZIP, GZIP, and Brotli showed similar, almost negligible decompression times for small dataset sizes. However, as the dataset size increased, the differences in decompression time became slightly more noticeable and followed a linear pattern. LZ4 consistently remained the fastest in compressing the data. Conversely, LZMA exhibited the slowest decompression speed, with noticeably high RSE values, followed by 7-Zip.

Fig 13 depicted the relationship between the bundle's content and the decompression time. A simple linear regression was used to investigate the correlation between the number of vertices and video size with decompression time.

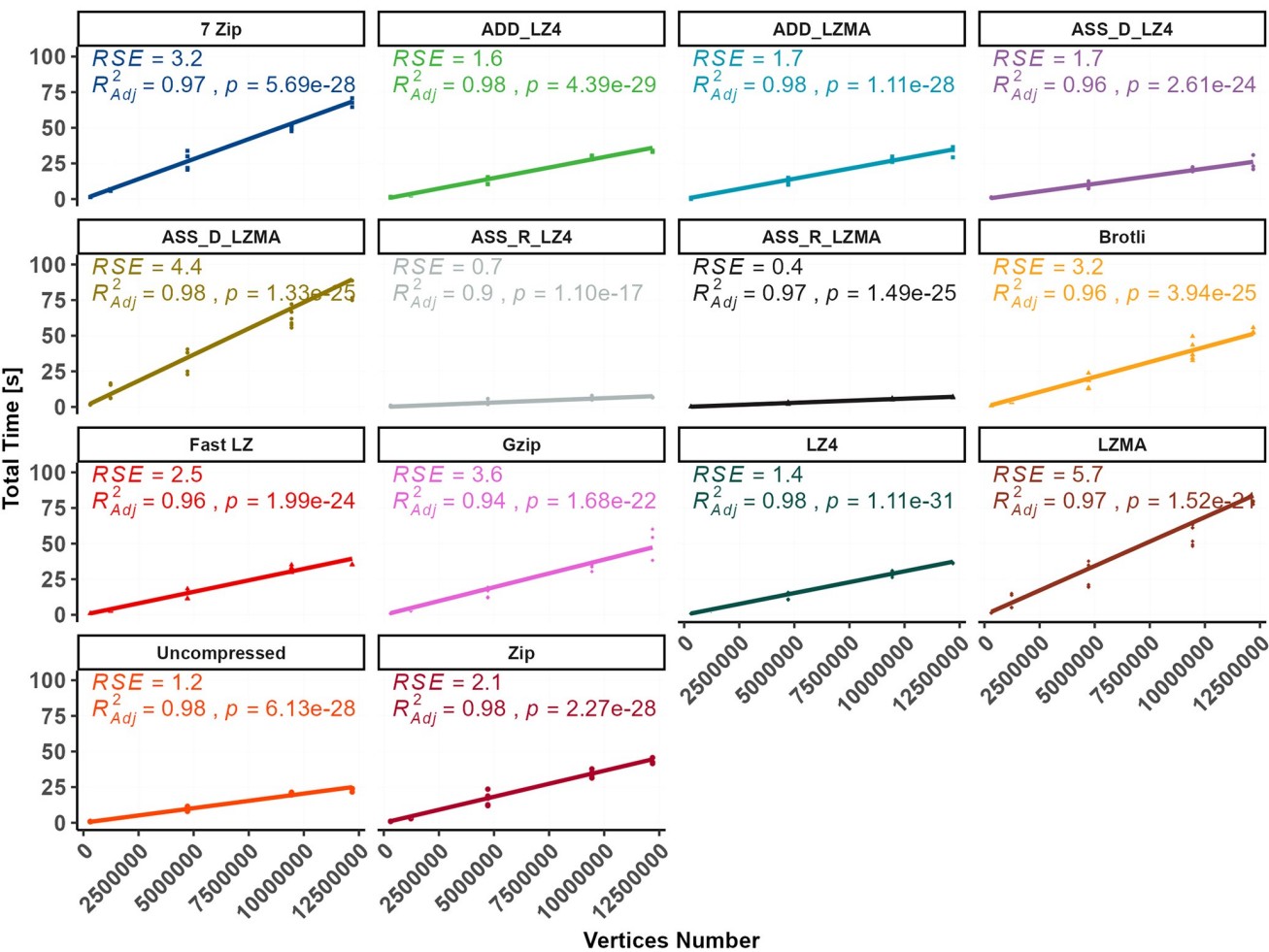

**Fig 9. Multiple linear regression of total time with vertices count and video size as predictors.**

A multiple linear regression model was developed to predict the time required for decompressing archived uncompressed bundles containing a 3D model and MP4 videos.

The equations displayed on the plot represented the best-fitted lines for each compression algorithm, with the models showing strong Adjusted R squared values and low RSE values. Table 8 provides additional statistical values and terms. LZMA is known for its high compression ratios, achieved through a relatively complex compression algorithm using dictionary-based compression and statistical modeling. Although this complexity results in excellent compression ratios, it can slow decompression. During compression, LZMA employs a sliding dictionary to represent repeated patterns efficiently, leading to higher compression ratios. However, during decompression, reconstructing the original data using the compressed file and dictionary becomes more time-consuming compared to more straightforward decompression methods. Additionally, the comparison of RSE values, as illustrated in Fig 14, reveals a significantly higher error for the LZMA compression method than other methods.

In this study, 7-Zip utilized LZMA2, which is generally faster at decompression than the original LZMA algorithm. LZMA2 achieves this by dividing the compressed data into independently decompressible blocks, enabling better utilization of multi-core processors, especially on modern CPUs with multiple cores. On the other hand, the original LZMA algorithm

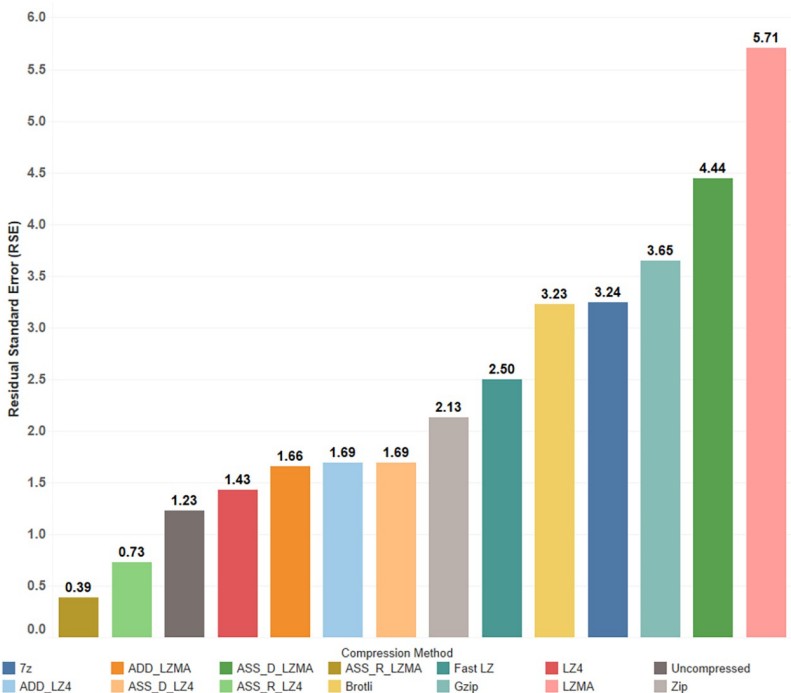

**Fig 10. Comparison of RSE values for total time regression with vertices count and video size.**

follows a more sequential process. It lacks the same level of parallelization, resulting in relatively slower decompression speeds, particularly on multi-core systems. Meanwhile, LZ4 and Fast LZ utilize smaller dictionaries and support vectorized instructions like SSE and AVX, contributing to faster decompression speeds at the cost of slightly lower compression ratios. These algorithms prioritize speed, overachieving the highest compression efficiency, making them suitable for scenarios where quick decompression is crucial, such as real-time data processing and network communication. Fig 15 depicts the decompression relationship in comparison to the number of vertices and video size.

## 4.4 RAM consumption

In this study, we have observed a gradual increase in RAM usage when employing specific asset bundle methods. To delve deeper into the correlation between these methods, we present Fig 16, illustrating the relationship.

It is noteworthy that all compression methods exhibit a strong and statistically significant linear relationship with the uncompressed bundle's maximum required RAM. The proportion of the variance in RAM usage ranges between 0.75 and 0.99 for these methods.

Notably, the adjusted R-squared values reveal higher coefficients for specific methods. Specifically, the RAM caching method using LZMA and LZ4, as well as 7-Zip and ASS_D_LZMA, attain an impressive, adjusted R-squared value of 0.99. In contrast, all other compression algorithms achieve an adjusted R-squared value of 0.75. Moreover, when exploring the correlation with the number of vertices, using external libraries such as Brotli, LZ4, LZMA, Fast LZ, Zip, and Gzip, along with uncompressed methods, ADD_LZ4, and ADD_LZMA, evidence a stronger correlation compared to other approaches.

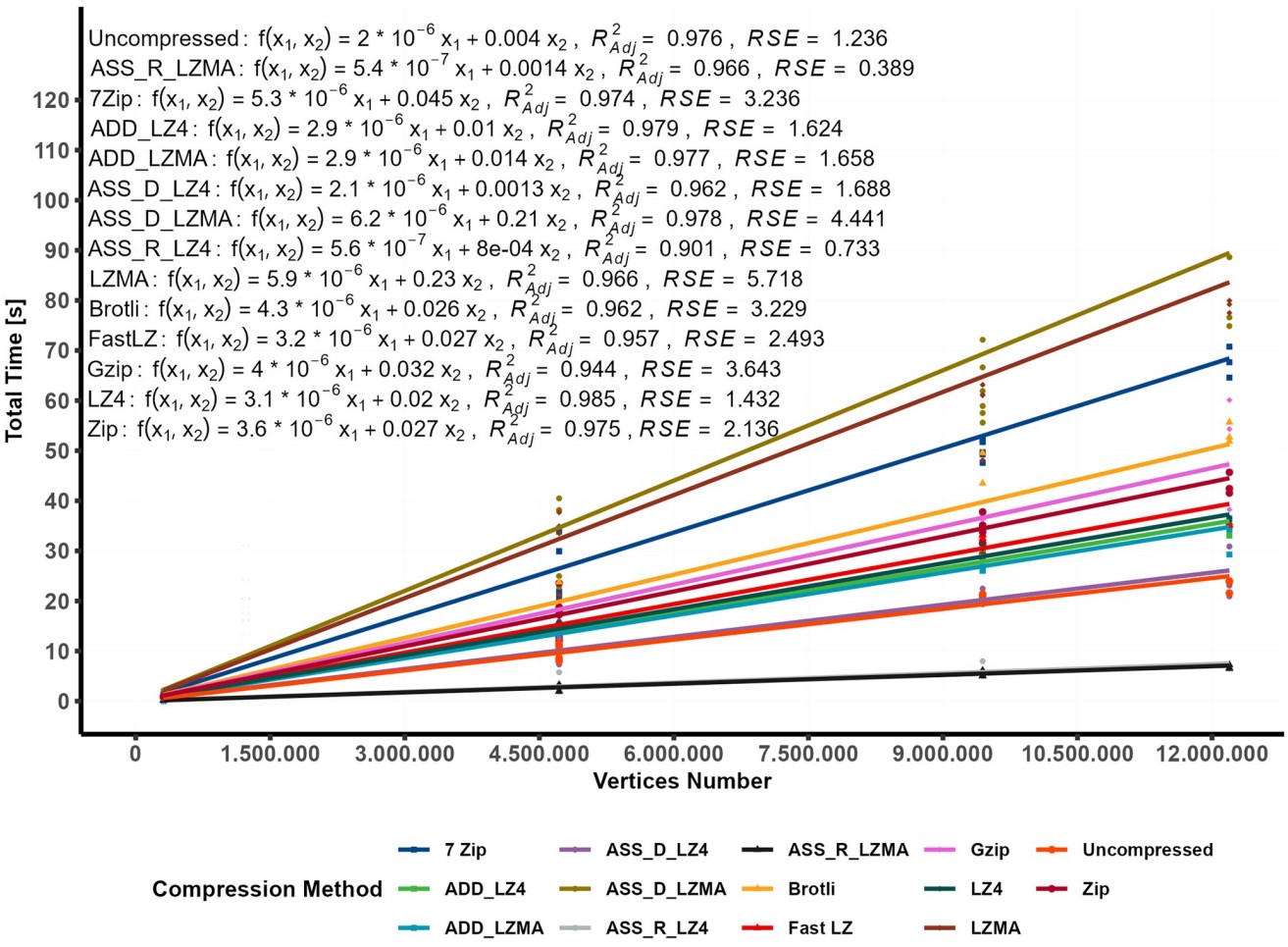

**Fig 11. Modeling total time using vertices count and video size as predictors (linear regression).**

Fig 17 presents the maximum RAM usage observed across different asset bundle methods. Surprisingly, the content type of the bundle does not seem to impact RAM consumption significantly when using the RAM caching method. Specifically, the asset bundle methods employed for caching the bundle in RAM, such as ASS_R_LZ4 and ASS_R_LZMA, exhibit a steady and consistent linear increase in RAM usage. The RAM usage associated with the asset bundle disk caching method using LZMA is considered the highest among all other compression methods.

The 7-Zip method demonstrates a stable and consistent linear increase in RAM usage, irrespective of the bundle's content. On the other hand, the remaining compression algorithms display slight variations in RAM usage with fluctuating behavior, which can be attributed to the diverse contents of the bundles. Based on the previous analysis, the data obtained from the different compression algorithms can be effectively fitted into multiple linear regression equations. Notably, since the asset bundles contain both 3D models and videos, the size of the video component is incorporated into the simple linear model. This integration of the video size into the model results in a slight improvement in both the RSE rate and the R-squared values for some models, leading to more unified and comprehensive multiple linear models for all compression methods. The multiple linear regression model accounts for the influence of

**Table 7. Summary of multiple linear regression analysis for total time prediction: Effects of vertex count and video size.**

| | Vertices | Confidence Interval | Video | Confidence Interval | R$^2$ Adj. | AIC | BIC | RSE |
|---|---|---|---|---|---|---|---|---|
| **LZMA** | 5.9e-06*** | [5.4e-06, 6.4e-06] | 0.23*** | [0.21, 0.24] | 0.966 | 232.6 | 238.9 | 5.71 |
| **LZ4** | 3.1e-06*** | [2.9e-06, 3.2e-06] | 0.02*** | [0.016, 0.024] | 0.985 | 132.9 | 139.2 | 1.43 |
| **Zip** | 3.6e-06*** | [3.4e-06, 3.8e-06] | 0.027*** | [0.021, 0.032] | 0.975 | 161.7 | 168 | 2.13 |
| **7-Zip** | 5.3e-06*** | [5e-06, 5.6e-06] | 0.045*** | [0.036, 0.054] | 0.974 | 191.6 | 197.9 | 3.24 |
| **Brotli** | 4.3e-06*** | [4e-06, 4.6e-06] | 0.026*** | [0.017, 0.034] | 0.962 | 191.4 | 197.8 | 3.23 |
| **Fast LZ** | 3.2e-06*** | [2.9e-06, 3.4e-06] | 0.027*** | [0.02, 0.034] | 0.957 | 172.8 | 179.1 | 2.50 |
| **Gzip** | 4e-06*** | [3.7e-06, 4.4e-06] | 0.032*** | [0.022, 0.042] | 0.944 | 200.1 | 206.4 | 3.65 |
| **ADD_LZMA** | 2.9e-06*** | [2.7e-06, 3e-06] | 0.014*** | [0.0098, 0.019] | 0.977 | 143.4 | 149.8 | 1.66 |
| **ADD_LZ4** | 2.9e-06*** | [2.7e-06, 3e-06] | 0.01*** | [0.0055, 0.014] | 0.979 | 141.9 | 148.3 | 1.62 |
| **ASS_D_LZMA** | 6.2e-06*** | [5.7e-06, 6.6e-06] | 0.21*** | [0.2, 0.22] | 0.978 | 214.4 | 220.7 | 4.44 |
| **ASS_D_LZ4** | 2.1e-06*** | [2e-06, 2.3e-06] | 0.0013 | [-0.0033, 0.006] | 0.962 | 144.7 | 151.1 | 1.69 |
| **ASS_R_LZMA** | 5.4e-07*** | [5e-07, 5.7e-07] | 0.0014* | [3e-04, 0.0024] | 0.966 | 39.1 | 45.5 | 0.39 |
| **ASS_R_LZ4** | 5.6e-07*** | [5e-07, 6.3e-07] | 0.00081 | [-0.0012, 0.0028] | 0.901 | 84.7 | 91 | 0.73 |
| **Uncomp.** | 2e-06*** | [1.9e-06, 2.1e-06] | 0.0041* | [0.00069, 0.0075] | 0.976 | 122.3 | 128.6 | 1.23 |

+ p<0.1,

* p<0.05,

** p<0.01,

*** p<0.001

both 3D models and videos on RAM usage by including the video size as an additional independent variable. The unified multiple linear model enhances the accuracy of predictions and provides a more complete picture of the relationship between RAM usage and the composition of the asset bundles.

An exciting observation emerges as we analyze the RAM usage across different compression algorithms in relation to the increasing number of vertices. Most compression algorithms exhibit a similar steady linear increase in RAM usage as the vertices number grows.

To gain a deeper understanding of the RAM usage patterns for each compression method, we have modeled the data, as depicted in Fig 18 and Table 9.

The modeling enables us to capture the trends and relationships within the dataset effectively. A comparison of the RSE values for the model is illustrated in Fig 19. Notably, the maximum PE_RSE is ca. 7%, indicating high accuracy in our modeling approach. These findings provide valuable insights into the RAM consumption patterns of various compression algorithms, shedding light on their efficiency and performance characteristics in handling datasets with varying numbers of vertices. By accurately modeling the RAM usage, we offer developers and researchers a comprehensive understanding of how these algorithms behave under different scenarios. This knowledge is crucial for making informed decisions when selecting the most appropriate compression method for specific applications, optimizing resource allocation, and ensuring efficient performance in resource-intensive systems. The analysis of RAM consumption further unveiled a common trend among all compression algorithms, showcasing steady and consistent linear increments in RAM usage. Remarkably, this behavior remained consistent irrespective of the bundle's content. Notably, the disparity in RAM consumption between the RAM and Disk caching methods widens as the bundle size increases. In applications where the bundle size is modest and rapid display times are paramount, opting for the RAM caching method is advisable. On the other hand, for larger datasets and scenarios with constrained download speeds, the Disk caching method might be more appropriate. This

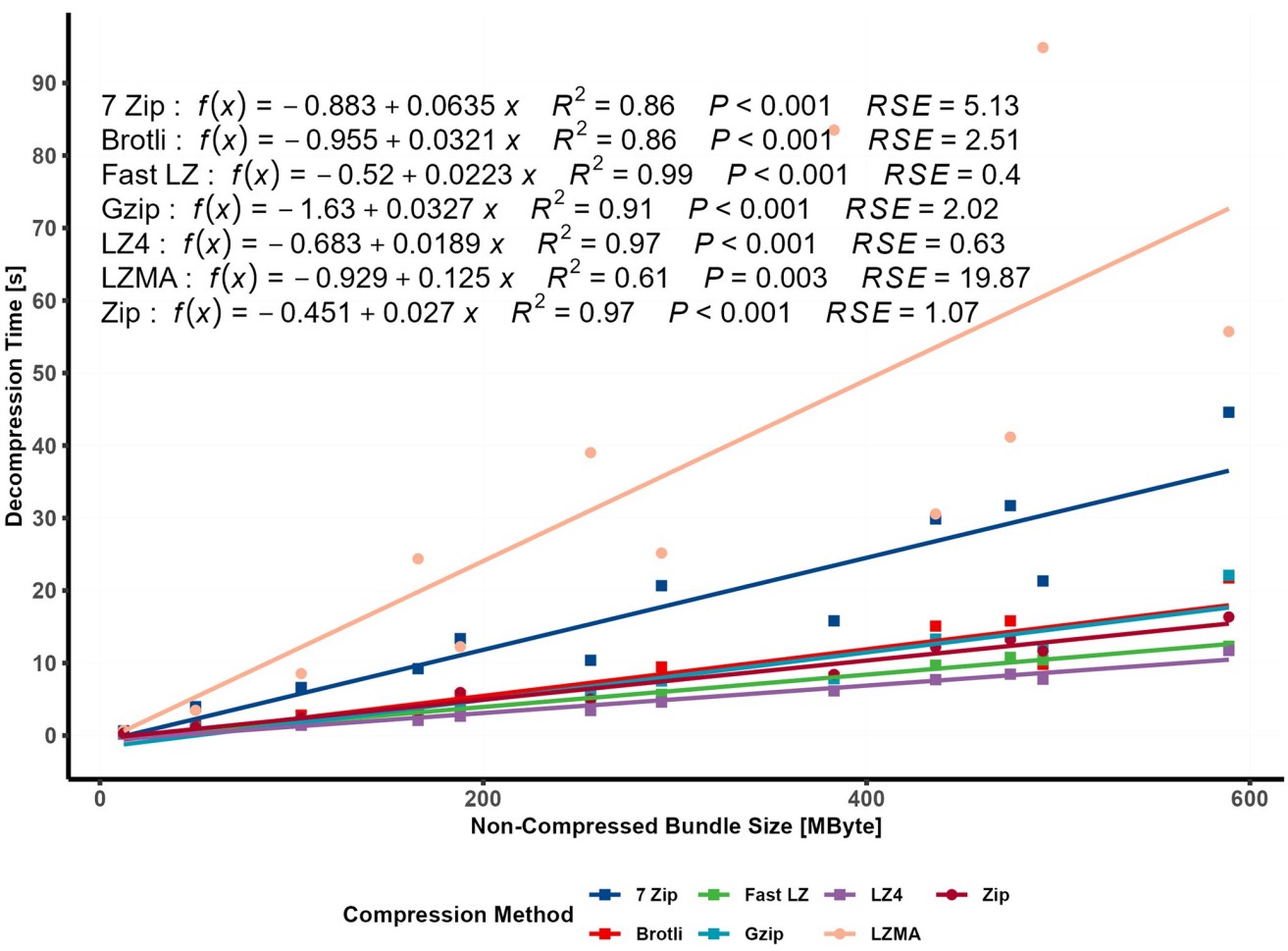

**Fig 12. Relationship between decompression time and bundle size (linear regression).**

discernment allows for tailored caching strategies based on specific project demands, ensuring optimal memory utilization and overall performance.

## 5 Conclusion and future directions

The study encapsulates an array of analyses, from RAM consumption to decompression speed, providing comprehensive insights into the performance dynamics of diverse compression algorithms. It underscores the significance of considering data structure, content, and characteristics when selecting appropriate compression methods for specific applications. The conclusions drawn from this comprehensive study serve as a roadmap for informed decision-making in the realm of data compression algorithms, paving the way for more efficient and optimized systems.

In terms of advantages and disadvantages, specific asset bundle methods were found to exhibit efficiency and stability, particularly those using RAM caching, such as ASS_R_LZ4 and ASS_R_LZMA, as well as the disk caching method with LZMA. RAM caching with LZ4 proved to be the fastest method for displaying assets due to its ability to access and manage data quickly in memory, making it highly suitable for performance-critical applications.

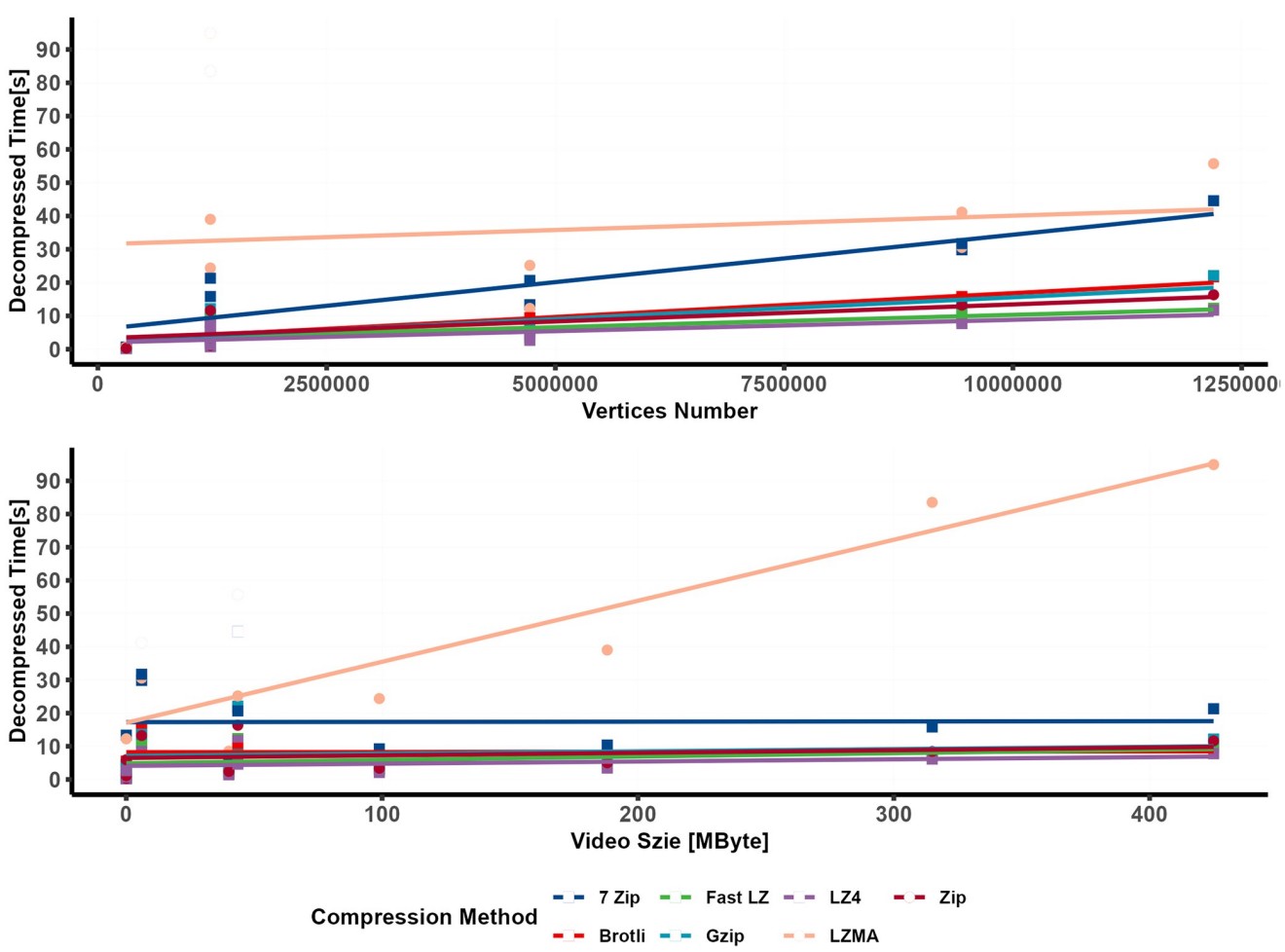

**Fig 13. Decompression time in relation to vertices count and video size (linear regression).**

**Table 8. Summary of multiple regression for decompression time prediction: Effects of vertex count and video size.**

|  | Vertices | Confidence Interval | Video | Confidence Interval | R² Adj. | AIC | BIC | RSE |
|---|---|---|---|---|---|---|---|---|
| **LZMA** | 4e-06*** | [3.1e-06, 4.9e-06] | 0.23*** | [0.2, 0.26] | 0.972 | 77.3 | 79.3 | 5.01 |
| **LZ4** | 9.2e-07*** | [8.3e-07, 1e-06] | 0.017*** | [0.015, 0.02] | 0.982 | 21.2 | 23.2 | 0.48 |
| **Zip** | 1.3e-06*** | [1.2e-06, 1.4e-06] | 0.023*** | [0.021, 0.026] | 0.992 | 19.9 | 21.5 | 0.48 |
| **7-Zip** | 3.4e-06*** | [3.1e-06, 3.7e-06] | 0.04*** | [0.031, 0.049] | 0.981 | 52.3 | 54.2 | 1.77 |
| **Brotli** | 1.7e-06*** | [1.6e-06, 1.9e-06] | 0.02*** | [0.016, 0.025] | 0.983 | 34.4 | 36.4 | 0.84 |
| **Fast LZ** | 1e-06*** | [9.2e-07, 1.2e-06] | 0.022*** | [0.019, 0.026] | 0.977 | 28 | 30 | 0.65 |
| **Gzip** | 1.7e-06*** | [1.4e-06, 1.9e-06] | 0.026*** | [0.02, 0.033] | 0.963 | 43.9 | 45.9 | 1.25 |

+ $p<0.1$,

* $p<0.05$,

** $p<0.01$,

*** $p<0.001$

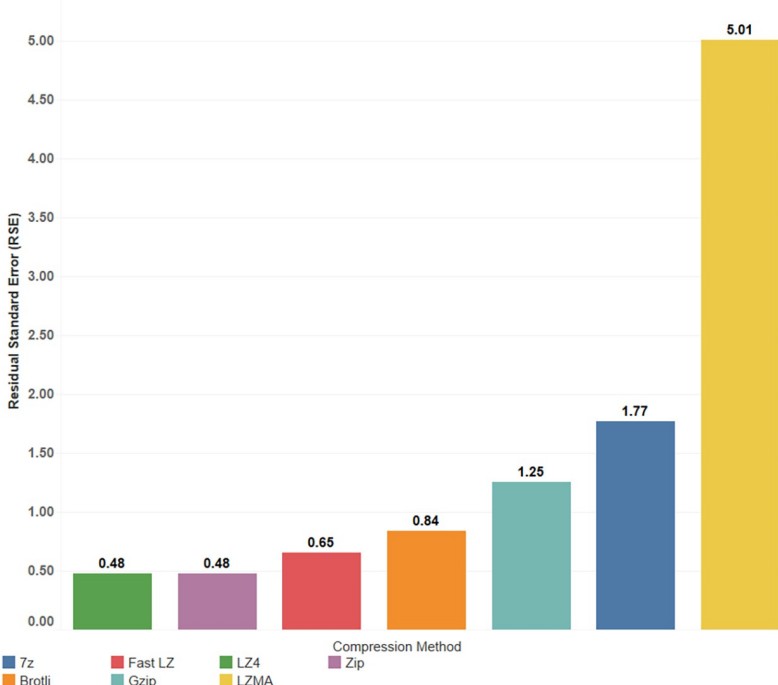

**Fig 14. Comparison of RSE values for decompression time regression with vertices count and video size.**

However, while LZMA offered the highest compression ratios, it suffered from significantly slower decompression times, making it less ideal for real-time applications where speed is crucial.

In contrast, Fast LZ and LZ4 consistently generated lower but stable compression ratios. These algorithms excel in scenarios where fast decompression times are prioritized over the highest compression efficiency. 7-Zip/LZMA2, while yielding the highest compression ratios, comes with the trade-off of slower decompression speeds, particularly for larger datasets. Brotli, when used in WebGL applications, provided a favorable balance with superior compression ratios compared to Gzip without compromising on decompression speed. This makes Brotli a strong candidate for web-based AR/VR content where both compression efficiency and speed are critical.

When examining 3D model characteristics, the relationship between vertices count and polygon count with compressed file size was scrutinized. Both variables substantially influenced compression size, as evident from high R-squared values. The Residual Standard Error (RSE) analysis revealed that vertice count was a more effective predictor than polygon count across all compression algorithms. Notably, 7-Zip stood out, exhibiting lower RSE error percentages when utilizing polygons as a predictor, but this came at the cost of higher decompression times.

In summary, the pros and cons of the various algorithms reveal distinct trade-offs: algorithms like LZ4 and Fast LZ prioritize speed over maximum compression efficiency, making them suitable for real-time applications, while LZMA and 7-Zip offer superior compression ratios but are slower in decompression, making them more suitable for storage or less time-

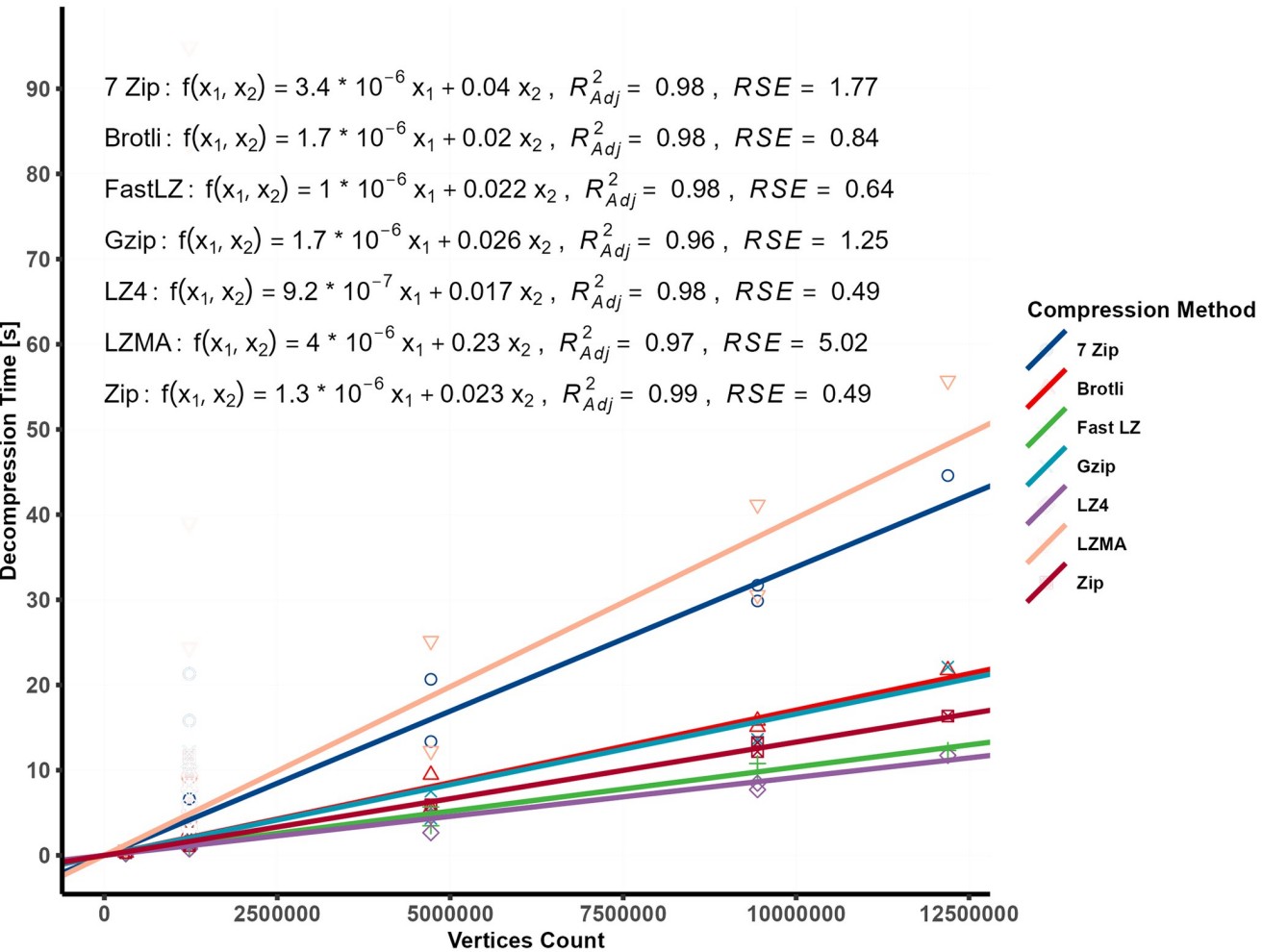

**Fig 15. Decompression time modeled compared to both vertices and video size.**

sensitive tasks. Brotli emerges as a strong performer in web applications due to its balanced performance, offering both high compression ratios and relatively fast decompression speeds.

Despite these findings, several limitations remain, opening up future research opportunities. While our study focused on popular algorithms such as LZ4, LZMA, and 7-Zip, future research could extend to other compression techniques such as Zstd, which is known for its speed and good compression ratios; Snappy, optimized for very high-speed compression and decompression; and LZHAM, a high-ratio alternative to LZMA with faster decompression. Another area worth exploring is cloud-based testing environments. Our research utilized local FTP-based testing, but future studies could simulate real-world scenarios where assets are stored and accessed via cloud services. This would provide insights into how compression algorithms perform under network latency and bandwidth constraints, which are critical for mobile and web-based applications.

Moreover, investigating live decompression techniques, where assets are decompressed on the fly during streaming, could be particularly beneficial for applications with dynamic content or live updates, enhancing the user experience without significant delays. Another promising

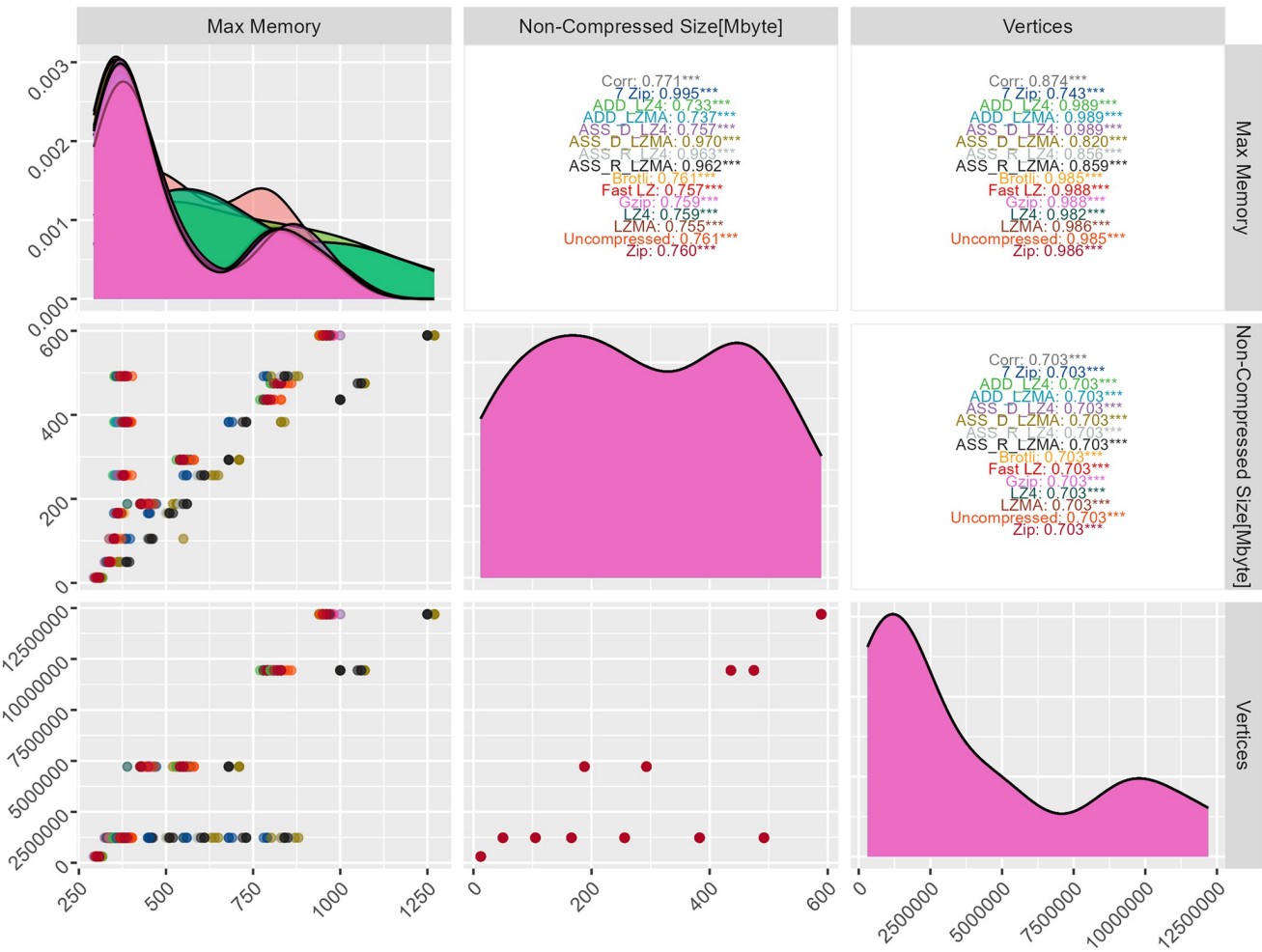

**Fig 16. Correlation analysis: Relationships among RAM, bundle size, and vertices count.**

direction is exploring real-time adaptive compression techniques, which dynamically select the most appropriate compression algorithm based on the current device state, network conditions, and user interaction patterns. This could optimize both performance and resource utilization. Advanced compression methods, such as neural compression or context-based algorithms, also warrant further examination. These approaches could potentially offer better performance for specific data types, especially in resource-constrained environments like mobile or AR/VR applications.

Furthermore, integrating these compression strategies with emerging technologies such as 5G and edge computing could enhance performance by offloading decompression tasks to more powerful edge servers. This would reduce latency and improve real-time performance in applications that require frequent updates or large data transfers. By addressing these limitations and exploring the outlined future research directions, we can further optimize data compression strategies for modern applications, particularly in resource-constrained environments like AR/VR on mobile devices. These efforts will continue to drive the efficiency of digital twin and immersive technology applications, enabling more robust and scalable systems.

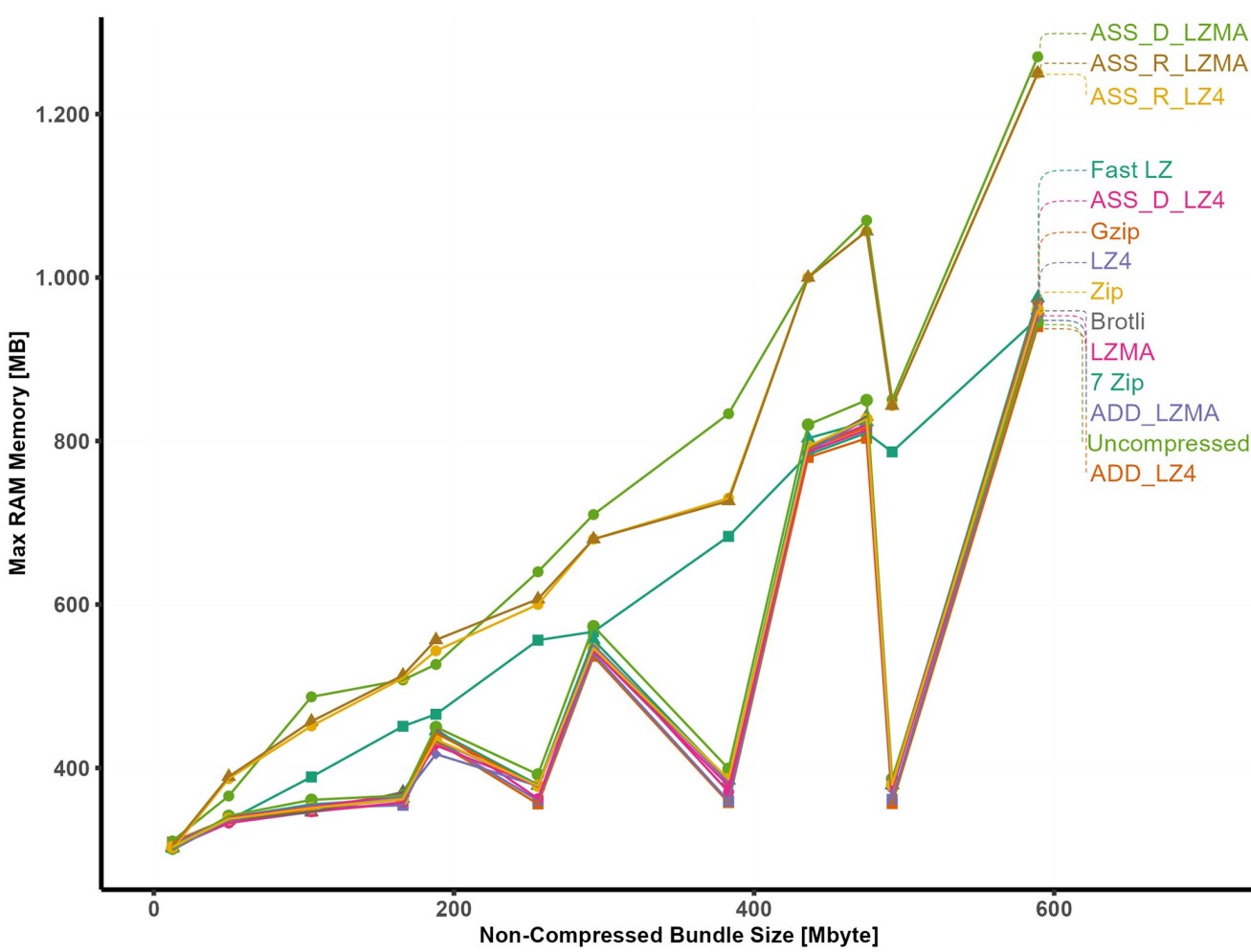

**Fig 17. RAM usage in relation to non-compressed bundle size.**

## 6 Recommendation

The analysis of RAM consumption further unveiled a common trend among all compression algorithms, showcasing steady and consistent linear increments in RAM usage. Remarkably, this behavior remained consistent irrespective of the bundle's content. Notably, the disparity in RAM consumption between the RAM and Disk caching methods widens as the bundle size increases. In applications where the bundle size is modest and rapid display times are paramount, opting for the RAM caching method is advisable. On the other hand, for larger datasets and scenarios with constrained download speeds, the Disk caching method might be more appropriate. This discernment allows for tailored caching strategies based on specific project demands, ensuring optimal memory utilization and overall performance.

The utilization of the AssetBundle method coupled with LZMA compression exhibited fluctuating outcomes, potentially stemming from the employment of the LoadFromFile(Async) approach. This method tends to consume more memory than alternatives like LoadFromCacheOrDownload(). Conversely, observations indicated that decompressing archived uncompressed bundles using external compression engines led to notably enhanced CPU frame rates by employing multithreading methodologies. Though garbage collection is essential for

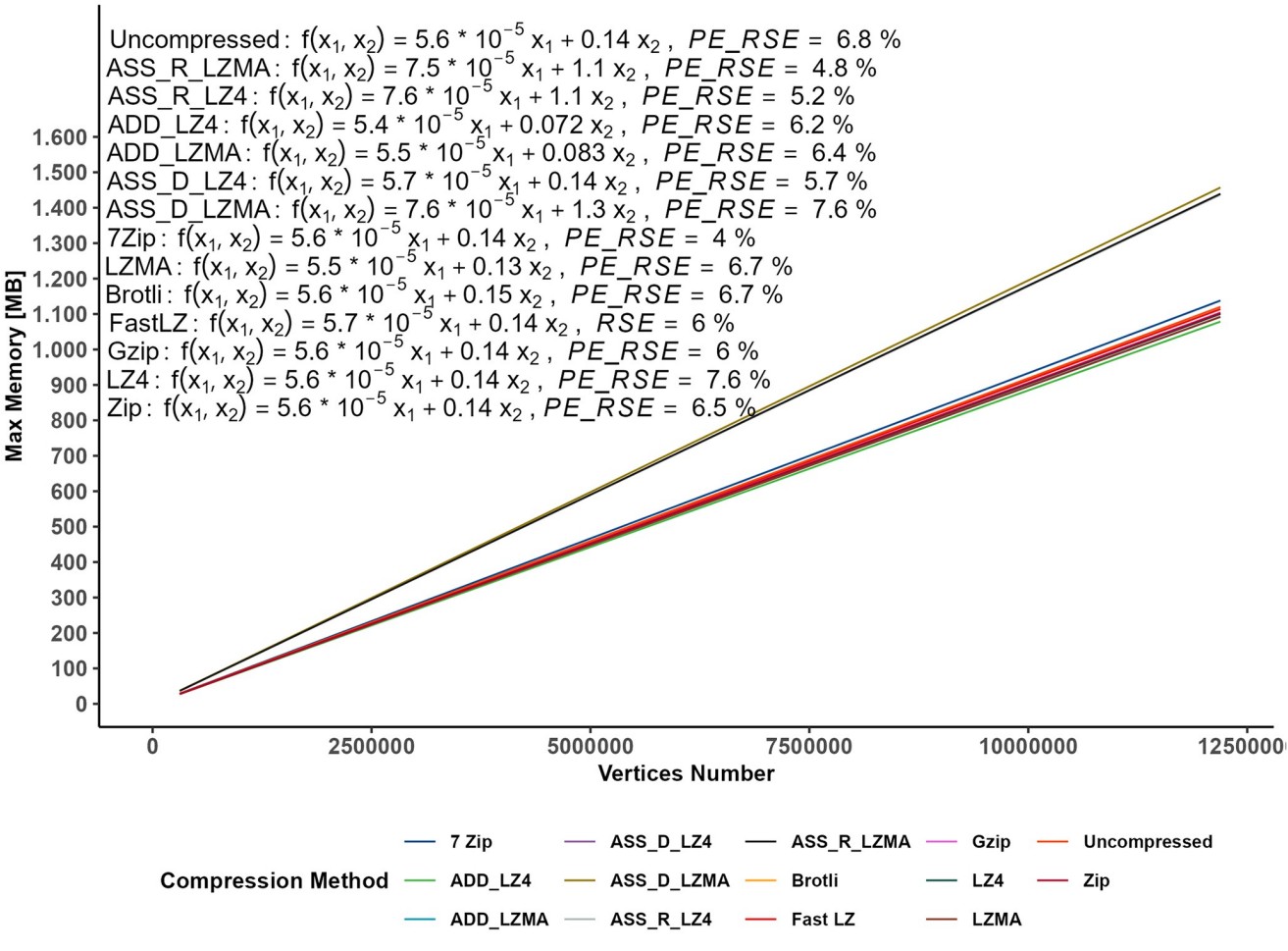

**Fig 18. Maximum RAM utilization in relation to vertices count and video size.**

memory management, its execution can influence the CPU frame rate. Diligent optimization and adept memory management techniques are crucial in mitigating this impact, ensuring a smoother and more consistent frame rate across Unity applications.

Furthermore, a noteworthy observation was made regarding utilizing the Addressable Asset Management for the decompression and presentation of Gameobjects, showcasing superior performance compared to the conventional Assetbundle tool. Irrespective of the compression method, the time needed for bundle decompression and GameObject loading proved significantly shorter with the Addressable Asset Management approach. This approach facilitated accelerated processes and maintained a steady and consistent RAM and CPU frame rate throughout the download and visualization phases.

## Acknowledgments

The authors express their gratitude to the Faculty of Computer Science and Information Technology (FSKTM) at Universiti Tun Hussein Onn Malaysia (UTHM) and the Fatima College of Health Sciences (FCHS) at the Institute of Applied Technology (IAT) for generously providing the necessary facilities.

**Table 9. Summary of multiple linear regression analysis for maximum RAM prediction: Effects of vertex count and video size.**

|  | Vertices | Confidence Interval | Video | Confidence Interval | R² Adj. | AIC | BIC | RSE |
|---|---|---|---|---|---|---|---|---|
| **LZMA** | 5.5e-05*** | [5.2e-05, 5.8e-05] | 0.13** | [0.037, 0.22] | 0.976 | 360 | 366 | 33.65 |
| **LZ4** | 5.6e-05*** | [5.2e-05, 6e-05] | 0.14** | [0.039, 0.25] | 0.970 | 369 | 376 | 38.43 |
| **ADD_LZMA** | 5.5e-05*** | [5.2e-05, 5.8e-05] | 0.083 | [-0.04, 0.17] | 0.979 | 355 | 362 | 31.61 |
| **ADD_LZ4** | 5.4e-05*** | [5.1e-05, 5.7e-05] | 0.072 | [-0.01, 0.16] | 0.979 | 353 | 360 | 30.74 |
| **Zip** | 5.6e-05*** | [5.3e-05, 5.9e-05] | 0.14** | [0.054, 0.23] | 0.978 | 358 | 364 | 32.76 |
| **7-Zip** | 5.2e-05*** | [5e-05, 5.4e-05] | 1.1*** | [1, 1.1] | 0.988 | 329 | 336 | 22.05 |
| **Brotli** | 5.6e-05*** | [5.3e-05, 5.9e-05] | 0.15** | [0.054, 0.24] | 0.977 | 360 | 366 | 33.57 |
| **Fast LZ** | 5.7e-05*** | [5.4e-05, 5.9e-05] | 0.14** | [0.054, 0.22] | 0.981 | 352 | 359 | 30.41 |
| **Gzip** | 5.6e-05*** | [5.3e-05, 5.9e-05] | 0.14** | [0.057, 0.23] | 0.981 | 343 | 349 | 30.24 |
| **ASS_D_LZMA** | 7.6e-05*** | [7.1e-05, 8.1e-05] | 1.3*** | [1.1, 1.4] | 0.965 | 394 | 400 | 54.23 |
| **ASS_D_LZ4** | 5.7e-05*** | [5.4e-05, 5.9e-05] | 0.14** | [0.058, 0.22] | 0.983 | 348 | 355 | 28.74 |
| **ASS_R_LZMA** | 7.5e-05*** | [7.2e-05, 7.8e-05] | 1.1*** | [1, 1.2] | 0.985 | 360 | 366 | 33.81 |
| **ASS_R_LZ4** | 7.6e-05*** | [7.2e-05, 7.9e-05] | 1.1*** | [1, 1.2] | 0.984 | 365 | 371 | 36.2 |
| **Uncompressed** | 5.6e-05*** | [5.3e-05, 5.9e-05] | 0.14** | [0.046, 0.24] | 0.975 | 362 | 369 | 34.84 |

+ p<0.1,

* p<0.05,

** p<0.01,

*** p<0.001

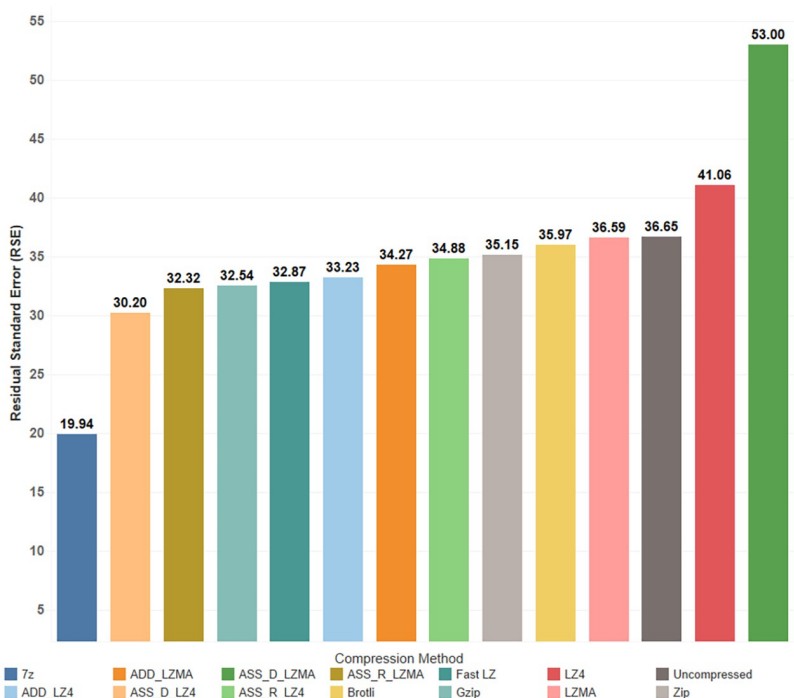

**Fig 19. RSE comparison for decompression time regression with vertices count and video size.**

## Author Contributions

**Conceptualization:** Mohammed Hlayel.

**Data curation:** Mohammed Hlayel, Mohammad Hayajneh, Siti Salwani Yaacob.

**Formal analysis:** Mohammed Hlayel, Hairulnizam Mahdin, Mohammad Hayajneh.

**Funding acquisition:** Mohammed Hlayel.

**Investigation:** Mohammed Hlayel, Hairulnizam Mahdin, Siti Salwani Yaacob, Mazidah Mat Rejab.

**Methodology:** Mohammed Hlayel.

**Project administration:** Mohammed Hlayel, Hairulnizam Mahdin.

**Resources:** Mohammed Hlayel, Mazidah Mat Rejab.

**Software:** Mohammed Hlayel, Saleh H. AlDaajeh.

**Supervision:** Hairulnizam Mahdin.

**Validation:** Mohammad Hayajneh.

**Visualization:** Mohammed Hlayel, Saleh H. AlDaajeh.

**Writing – original draft:** Mohammed Hlayel.

**Writing – review & editing:** Mohammed Hlayel, Hairulnizam Mahdin, Mazidah Mat Rejab.

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
