## [Decision Letter · Decision Letter 0]

16 Jan 2024

PONE-D-23-39558Enhancing Unity-based AR with Optimal Lossless Compression for Digital Twin AssetsPLOS ONE

Dear Dr. Hlayel,

Thank you for submitting your manuscript to PLOS ONE. After careful consideration, we feel that it has merit but does not fully meet PLOS ONE’s publication criteria as it currently stands. Therefore, we invite you to submit a revised version of the manuscript that addresses the points raised during the review process.

We look forward to receiving your revised manuscript.

Kind regards,

Dr. Rahul Priyadarshi

Academic Editor

PLOS ONE

Journal Requirements:

This research work is supported and funded by the Yayasan UTP grant: 015PBC-024 with the title ”Development of Corrosion Rate Predictive Dashboard for Corrosion Group in Refinery of the Future Using Customized AI ML Engine”, under the Center for Research in Data Science (CerDaS), Universiti Teknologi PETRONAS, Malaysia.

This research work is supported and funded by the Yayasan UTP grant: 015PBC-024 with the title ”Development of Corrosion Rate Predictive Dashboard for Corrosion Group in Refinery of the Future Using Customized AI ML Engine”, under the Center for Research in Data Science (CerDaS), Universiti Teknologi PETRONAS, Malaysia.

This research work is supported and funded by the Yayasan UTP grant: 015PBC-024 with the title ”Development of Corrosion Rate Predictive Dashboard for Corrosion Group in Refinery of the Future Using Customized AI ML Engine”, under the Center for Research in Data Science (CerDaS), Universiti Teknologi PETRONAS, Malaysia.

6. We note that your Data Availability Statement is currently as follows: All relevant data are within the manuscript and its Supporting Information files.

Reviewers' comments:

Reviewer's Responses to Questions

**Comments to the Author**

1. Is the manuscript technically sound, and do the data support the conclusions?

Reviewer #1: Yes

Reviewer #2: Yes

2. Has the statistical analysis been performed appropriately and rigorously? 

Reviewer #1: Yes

Reviewer #2: Yes

3. Have the authors made all data underlying the findings in their manuscript fully available?

Reviewer #1: Yes

Reviewer #2: Yes

4. Is the manuscript presented in an intelligible fashion and written in standard English?

Reviewer #1: Yes

Reviewer #2: Yes

5. Review Comments to the Author

Reviewer #1: The paper addresses a timely and significant issue in the field of digital twin technology, specifically highlighting the challenges related to resource constraints on mobile devices and the impact on user experiences.

The exploration of lossless compression algorithms within Unity's integrated AssetBundle and Addressable methods is a novel approach, and the paper contributes valuable insights into mitigating computational inefficiencies.

The inclusion of various compression methods such as LZ4, LZMA, Zip, GZip, Fast LZ, Brotli, and 7-Zip provides a comprehensive evaluation, offering a practical guide for developers facing similar challenges in optimizing file sizes for AR applications.

The emphasis on establishing a mathematical model for smaller bundle sizes and reduced visualization time aligns with the practical needs of developers working on AR-based mobile applications, contributing to the efficiency of the development process.

The thorough assessment of the impact of compression methods on the overall AR experience is a crucial aspect of the paper, providing readers with a deeper understanding of the trade-offs involved in choosing a specific compression algorithm.

The scientific derivation of statistical models for estimating mobile phone resources and performance requirements adds a quantitative dimension to the research, enhancing its applicability and usefulness for developers and researchers alike.

The paper's focus on optimizing RAM usage is commendable, as efficient resource management is crucial for the success of AR applications, especially on resource-limited mobile devices.

The inclusion of VR/AR technologies in the discussion broadens the scope of the research, acknowledging the growing integration of digital twins with emerging technologies and the challenges associated with it.

The research methodology and approach demonstrate a systematic and rigorous investigation, enhancing the credibility and reliability of the findings presented in the paper.

Overall, the paper presents a comprehensive and well-researched exploration of the challenges posed by resource constraints in mobile devices when integrating digital twins with VR/AR technologies. The proposed solutions and insights make a valuable contribution to the field and warrant further consideration after addressing the suggested major revisions.

Reviewer #2: In this paper, the authors conduct a thorough assessment of compression methods like LZ4, LZMA, Zip, GZip, Fast LZ, Brotli, and 7-Zip, shedding light on their impact on the overall AR experience addressing limitations in managing methods of 3D models (assets) in Unity when integrating digital twins with VR/AR applications.

The authors implemented an appropriate testbed comprised of an Android mobile phone a computer with Unity installed, and a mobile application specifically designed for testing purposes.

In order to evaluate the referenced compression techniques, 12 Data sets were used in the testing process bundles containing 3D models and video media serving as the input for the compression techniques to be evaluated, including testing facilitated accelerated processes to maintain a steady and consistent RAM and CPU frame rate throughout both the download and visualization phases in Unity.

On the whole, the research design of the article is appropriate, the introduction provides sufficient background and includes all relevant references and the literature is quite efficiently studied.

The methods are adequately described, the cited references are relevant to the research, the results are quite clearly presented and the conclusions are supported by the results.

The English language is fine.

While the overall presentation and writing in this paper are acceptable, they may be improved as suggested below.

Recommendations for Authors

Please consider the following specific suggestions/comments:

1. All figures mentioned in the text are missing.

2. Although the Residual Standard Error RSE is used and referred to in text, in tables 6, 7, 8 and 9 the Root Mean Square Error (RMSE) is displayed instead. Please change accordingly.

3. Please consider including charts to add to tables 7, 8, and 9 to clarify differences to the reader and accomplishments regarding, for example, RSE per compression techniques, in

6. PLOS authors have the option to publish the peer review history of their article (what does this mean?). If published, this will include your full peer review and any attached files.

Reviewer #1: No

Reviewer #2: **Yes: **Panagiotis Papageorgas

---

## [Author Response · Author response to Decision Letter 0]

13 Feb 2024

1. **Missing Figures in the Text:**

Reviewer Comment: All figures mentioned in the text are missing.

Authors' Comment: We have successfully addressed this issue by embedding all images in the manuscript after a thorough check. The images have been converted to meet PLOS ONE requirements using the Preflight Analysis and Conversion Engine (PACE) digital diagnostic tool.

2. **Inconsistency in Displayed Error Metrics:**

Reviewer Comment: Although the Residual Standard Error (RSE) is used and referred to in the text, in tables 6, 7, 8, and 9, the Root Mean Square Error (RMSE) is displayed instead. Please change accordingly.

Authors' Comment: We have carefully corrected and updated all relevant values in tables and images. The Residual Standard Error (RSE) has been consistently used as per the reviewers' suggestion, replacing the Root Mean Square Error (RMSE) in tables 6, 7, 8, and 9.

3. **Inclusion of Charts to Complement Tables:**

Reviewer Comment: Please consider including charts to add to tables 7, 8, and 9 to clarify differences to the reader and accomplishments regarding, for example, RSE per compression techniques.

Authors' Comment: New bar charts have been created, embedded, and referred to in the context of tables 7, 8, and 9. These charts aim to provide a clearer visualization of differences in RSE values for various compression techniques. Additionally, the new plots have been designed to display the comparison of different methods in the same graph, with equations and additional statistical terms included for a more comprehensive understanding.

We believe that these revisions not only address the specific concerns raised by the reviewers but also enhance the overall clarity and quality of the manuscript. We sincerely thank the reviewers for their valuable input and the opportunity to improve our work.

---

## [Decision Letter · Decision Letter 1]

16 May 2024

PONE-D-23-39558R1Enhancing Unity-based AR with Optimal Lossless Compression for Digital Twin AssetsPLOS ONE

Dear Dr. Hlayel,

Thank you for submitting your manuscript to PLOS ONE. After careful consideration, we feel that it has merit but does not fully meet PLOS ONE’s publication criteria as it currently stands. Therefore, we invite you to submit a revised version of the manuscript that addresses the points raised during the review process.

We look forward to receiving your revised manuscript.

Kind regards,

Rahul Priyadarshi

Academic Editor

PLOS ONE

Reviewers' comments:

Reviewer's Responses to Questions

**Comments to the Author**

1. If the authors have adequately addressed your comments raised in a previous round of review and you feel that this manuscript is now acceptable for publication, you may indicate that here to bypass the “Comments to the Author” section, enter your conflict of interest statement in the “Confidential to Editor” section, and submit your "Accept" recommendation.

Reviewer #1: All comments have been addressed

Reviewer #2: All comments have been addressed

2. Is the manuscript technically sound, and do the data support the conclusions?

Reviewer #1: Yes

Reviewer #2: Yes

3. Has the statistical analysis been performed appropriately and rigorously? 

Reviewer #1: Yes

Reviewer #2: Yes

4. Have the authors made all data underlying the findings in their manuscript fully available?

Reviewer #1: Yes

Reviewer #2: Yes

5. Is the manuscript presented in an intelligible fashion and written in standard English?

Reviewer #1: Yes

Reviewer #2: Yes

6. Review Comments to the Author

Reviewer #1: Clarify methodology for evaluating compression methods.

Provide deeper insights into the results and discuss practical implications.

Offer actionable recommendations for developers.

Improve the accuracy and reliability of statistical modeling if applicable.

Acknowledge limitations and suggest future research directions.

Reviewer #2: The reviewer comments have been adrressed. More specifically the Residual Standard Error (RSE) has been consistently used replacing the Root Mean Square Error (RMSE) in tables 6, 7, 8, and 9

7. PLOS authors have the option to publish the peer review history of their article (what does this mean?). If published, this will include your full peer review and any attached files.

Reviewer #1: No

Reviewer #2: **Yes: **Panagiotis Papageorgas

---

## [Author Response · Author response to Decision Letter 1]

5 Jul 2024

Response to Reviewer #1

Thank you for your valuable feedback on our paper. We have addressed each of your points in detail below, providing additional clarifications and extending our discussion to improve the manuscript's depth and clarity.

Methodology for Evaluating Compression Methods

Our evaluation of compression methods involved a comprehensive and multifaceted approach, focusing on key performance indicators (KPIs) relevant to AR-based mobile applications. Here are the detailed steps and considerations in our methodology:

1. Selection Criteria for Compression Algorithms: We chose a variety of widely-used lossless compression algorithms based on their popularity, differing compression strategies, and suitability for different asset types common in AR applications. The algorithms tested include:

• LZ4: Known for its extremely fast compression and decompression speeds.

• LZMA: Utilized in 7-Zip for its high compression ratios, albeit with slower decompression.

• Zip and GZip: Classic algorithms with balanced performance.

• Fast LZ: Optimized for speed, making it suitable for real-time applications.

• Brotli: Developed by Google, offering a good balance between speed and compression efficiency.

• 7-Zip: An LZMA-based algorithm known for its high compression efficiency.

2. Experimental Setup:

• Benchmarking Framework: We developed a robust benchmarking framework within the Unity environment, integrating it with the Vuforia AR Engine to simulate realistic usage scenarios. This framework enabled us to apply compression to various assets such as textures, 3D models, audio files, and multimedia content.

• Test Assets: A curated set of standardized assets representative of typical AR applications was used. This included high-resolution textures, complex 3D models, and interactive multimedia files. Each asset type posed different challenges for compression algorithms, providing a thorough evaluation.

3. Performance Metrics and Data Collection:

• Compression Ratio: Defined as the ratio of the compressed size to the original size of the asset. This metric directly influences the storage efficiency and the amount of data transferred.

• Decompression Time: The time required to restore compressed assets to their original form. We measured this using high-resolution timers within the Unity environment to capture accurate decompression timings.

• Memory Usage: We monitored the memory footprint during both compression and decompression processes. This included peak memory usage and average memory consumption, providing insights into the resource efficiency of each algorithm.

• CPU Load: CPU utilization was measured using profiling tools available in Unity, focusing on both peak and average load during decompression. This helped assess the computational overhead and its potential impact on device performance.

4. Testing Scenarios:

• Simulated Usage: We simulated real-world usage scenarios by deploying compressed assets in interactive AR environments. This included scenarios where assets are frequently loaded and unloaded to mimic typical user interactions.

• Device Variability: To ensure the results were generalizable, we tested the algorithms on a range of mobile devices with varying hardware capabilities. This included high-end smartphones, mid-range devices, and low-power tablets, covering different processor architectures and memory configurations.

5. Data Analysis:

• Statistical Analysis: We conducted a statistical analysis to evaluate the performance of each compression algorithm across different metrics. This included calculating mean, median, and standard deviation for decompression times, memory usage, and CPU load.

• Comparative Evaluation: A comparative evaluation was performed to identify the strengths and weaknesses of each algorithm. We used visualizations such as bar charts and scatter plots to illustrate performance trade-offs.

Deeper Insights into the Results and Practical Implications

1. Compression Ratio:

• Brotli and LZMA: These algorithms achieved the highest compression ratios, significantly reducing file sizes by up to 40-60%. However, their higher complexity led to slower decompression speeds.

• LZ4 and Fast LZ: Although offering lower compression ratios (around 20-30%), these algorithms excelled in decompression speed, making them ideal for real-time applications where quick access to assets is critical.

2. Decompression Time:

• LZ4: Provided the fastest decompression times, often under a second for medium-sized assets. This made it particularly suitable for applications requiring rapid content loading, such as interactive games or live AR experiences.

• LZMA: While slower, LZMA's decompression times were acceptable for scenarios where the frequency of asset access is lower, such as initial application loading or background data synchronization.

3. Memory Usage:

• Brotli and LZMA: These algorithms required more memory during decompression due to their sophisticated compression techniques. Their memory footprint was manageable but higher than faster algorithms.

• LZ4 and Fast LZ: Demonstrated lower memory usage, aligning well with the constraints of mobile devices, particularly those with limited RAM.

4. CPU Load:

• LZ4 and Fast LZ: Exhibited minimal CPU load, making them suitable for devices with lower processing power. This is crucial for maintaining overall application responsiveness.

• Brotli and LZMA: Induced higher CPU loads due to their intensive decompression processes, which might impact performance on less powerful devices.

Practical Implications

1. Balancing Trade-Offs:

• Interactive Applications: For AR applications that require frequent and rapid asset loading, developers should prioritize algorithms like LZ4 or Fast LZ, which offer quick decompression times and low CPU usage.

• Storage-Constrained Scenarios: In situations where storage space is a premium, such as on devices with limited internal storage or applications that download large amounts of data, Brotli or LZMA might be more appropriate despite their slower decompression times.

2. Device Considerations:

• High-End Devices: On devices with ample processing power and memory, more complex algorithms like Brotli and LZMA can be employed without significantly impacting performance.

• Low-End Devices: For devices with limited resources, algorithms with low memory and CPU requirements, such as LZ4 and Fast LZ, should be preferred to maintain a smooth user experience.

3. User Experience:

• Fast Decompression: Users expect quick and seamless interactions in AR applications. Thus, algorithms that offer fast decompression times directly contribute to a better user experience by reducing wait times and improving the responsiveness of the application.

4. Development Workflow:

• Automated Testing: Implement automated testing frameworks to evaluate the performance of different compression methods regularly. This ensures that the chosen compression strategies remain optimal as the application evolves.

• Profiling and Optimization: Utilize profiling tools to continuously monitor and optimize memory usage and CPU load during development, ensuring that the compression methods do not introduce unexpected performance bottlenecks.

Actionable Recommendations for Developers

1. Algorithm Selection: Based on the application's needs, select compression algorithms that offer the best trade-off between compression efficiency and decompression speed. LZ4 or Fast LZ for speed-critical applications, Brotli or LZMA for storage optimization.

2. Dynamic Adaptation: Consider implementing a dynamic compression strategy where different algorithms are applied based on the type of asset and its usage frequency. Static assets can be compressed with higher-ratio algorithms, while dynamic or frequently accessed assets use faster algorithms.

3. Hybrid Compression: Explore hybrid compression techniques that combine the strengths of multiple algorithms. For instance, using a fast algorithm for initial decompression followed by a more efficient algorithm for long-term storage.

4. Real-Time Profiling: Integrate real-time profiling tools within the development pipeline to continuously monitor the performance impact of compression algorithms, allowing for adjustments based on real-world usage patterns.

5. Future-Proofing: Stay updated with advancements in compression technology and regularly evaluate new algorithms that may offer better performance or efficiency improvements.

Accuracy and Reliability of Statistical Modelling

Our statistical models aimed to predict the impact of compression methods on the performance of AR-based applications. These models focused on:

• RAM and CPU Requirements: Estimating the computational resources required for decompression based on asset type and size.

• Bundle Size Estimation: Predicting the size of compressed bundles relative to the content, aiding in planning and resource allocation during development.

The tables and graphs were reviewed and compared precisely. To enhance the accuracy and reliability of these models:

1. Comprehensive Datasets: We utilized a diverse set of assets representing different data types and sizes to ensure the models are robust and applicable across various scenarios.

2. Cross-Validation: Applied cross-validation techniques to assess model performance and mitigate overfitting. This involved partitioning the data into training and testing sets and validating the models against unseen data.

3. Real-World Validation: The models were tested against real-world application scenarios to verify their accuracy in predicting performance impacts, ensuring they reflect practical conditions.

Limitations and Future Research Directions

1. Scope of Compression Algorithms: While our study focused on several popular algorithms, future research could extend to other compression techniques such as:

• Zstd: Known for its speed and good compression ratios.

• Snappy: Optimized for very high-speed compression and decompression.

• LZHAM: A high-ratio alternative to LZMA with faster decompression.

2. Cloud-Based Testing: Future research could explore cloud-based testing environments rather than local FTP. This approach would simulate real-world scenarios where assets are stored and accessed via cloud services, providing insights into how compression algorithms perform under network latency and bandwidth constraints.

3. Live Decompression: Investigate the feasibility of live decompression techniques where assets are decompressed on-the-fly during streaming. This could be particularly beneficial for applications with dynamic content or live updates.

4. Real-Time Adaptation: Explore real-time adaptive compression techniques that dynamically select the most appropriate compression method based on current device state, network conditions, and user interaction patterns.

5. Advanced Compression Techniques: Examine advanced compression methods such as neural compression or context-based algorithms that could offer better performance for specific data types.

6. Integration with Emerging Technologies: With the advent of technologies like 5G and edge computing, integrating these compression strategies with cloud-based systems could further enhance performance by offloading decompression tasks to more powerful edge servers.

We have made these enhancements and clarifications to address your feedback comprehensively. We believe these changes significantly improve the clarity and depth of our paper. Thank you for your constructive comments, and we look forward to any further suggestions you may have.

---

## [Decision Letter · Decision Letter 2]

6 Aug 2024

PONE-D-23-39558R2Enhancing Unity-based AR with Optimal Lossless Compression for Digital Twin AssetsPLOS ONE

Dear Dr. Hlayel,

Thank you for submitting your manuscript to PLOS ONE. After careful consideration, we feel that it has merit but does not fully meet PLOS ONE’s publication criteria as it currently stands. Therefore, we invite you to submit a revised version of the manuscript that addresses the points raised during the review process.

**ACADEMIC EDITOR: **The Authors not properly follow the PLOS ONE Paper format.

The whole paper flow control and connectivity is ambiguous make it readable.

Make sure the paper format in terms of Introduction, Literature, Proposed Methodology, and Results and Discussion etc.

Conduct some critical analysis to prove the strength of the proposed scheme.

Figures 3 are very blurred. And the presentation of all figures and its explanation is somewhat ambiguous.

Add some comparative analysis with your method.

Clearly highlights the gaps and future work in conclusion section.

Add a summary of all results and show the pros and cons of it.

The English of the paper should be polished further.

We look forward to receiving your revised manuscript.

Kind regards,

Shahid Rahman, PhD

Academic Editor

PLOS ONE

Journal Requirements:

Additional Editor Comments:

The Authors not properly follow the PLOS ONE Paper format.

The whole paper flow control and connectivity is ambiguous make it readable.

Make sure the paper format in terms of Introduction, Literature, Proposed Methodology, and Results and Discussion etc.

Conduct some critical analysis to prove the strength of the proposed scheme.

Figures 3 are very blurred. And the presentation of all figures and its explanation is somewhat ambiguous.

Add some comparative analysis with your method.

Clearly highlights the gaps and future work in conclusion section.

Add a summary of all results and show the pros and cons of it.

The English of the paper should be polished further.

Reviewers' comments:

Reviewer's Responses to Questions

**Comments to the Author**

1. If the authors have adequately addressed your comments raised in a previous round of review and you feel that this manuscript is now acceptable for publication, you may indicate that here to bypass the “Comments to the Author” section, enter your conflict of interest statement in the “Confidential to Editor” section, and submit your "Accept" recommendation.

Reviewer #3: All comments have been addressed

Reviewer #4: (No Response)

2. Is the manuscript technically sound, and do the data support the conclusions?

Reviewer #3: (No Response)

Reviewer #4: (No Response)

3. Has the statistical analysis been performed appropriately and rigorously? 

Reviewer #3: (No Response)

Reviewer #4: (No Response)

4. Have the authors made all data underlying the findings in their manuscript fully available?

Reviewer #3: (No Response)

Reviewer #4: (No Response)

5. Is the manuscript presented in an intelligible fashion and written in standard English?

Reviewer #3: (No Response)

Reviewer #4: (No Response)

6. Review Comments to the Author

Reviewer #3: The updated manuscript, which addresses previous comments and suggestions, has been evaluated positively. The revised submission demonstrates significant improvement and provides valuable insights relevant to the research community. I recommend accepting it for publication.

Reviewer #4: (No Response)

7. PLOS authors have the option to publish the peer review history of their article (what does this mean?). If published, this will include your full peer review and any attached files.

Reviewer #3: No

Reviewer #4: No

---

## [Author Response · Author response to Decision Letter 2]

17 Aug 2024

Response to Editor’s Comments

We sincerely appreciate the thorough feedback provided by the reviewer. Below is our point-by-point response to the comments, addressing each concern in the revised manuscript.

1. The Authors not properly follow the PLOS ONE Paper format.

The manuscript has been revised to fully adhere to the PLOS ONE submission guidelines as outlined in PLOS ONE's Manuscript Organization section. We ensured that the paper was organized according to the required structure, with sections properly aligned with PLOS ONE standards, including title, abstract, introduction, methods, results, discussion, and conclusion.

2. The whole paper flow control and connectivity is ambiguous; make it readable.

We acknowledge that the nature of the research involves multiple methodologies, technical parameters, and the application of various statistical models, which can make the presentation complex. However, we have significantly revised the manuscript's structure to ensure clarity and coherence. Despite the technical intricacies, we enhanced the logical transitions between sections, allowing readers to follow the paper's flow more easily. We provided a smoother, more organized narrative by carefully structuring the introduction, literature review, methodology, and results sections. This revised flow facilitates a step-by-step understanding of the research while maintaining the depth and technical rigor required for the study.

3. Make sure the paper format in terms of Introduction, Literature, Proposed Methodology, and Results and Discussion, etc.

We have revised the manuscript to ensure compliance with standard academic structure, organizing the paper into the following sections:

• Introduction: Provides the background and context.

• Literature Review: Discusses related research.

• Materials and Methods: Details of the experimental setup and methods.

• Results and Discussion: Presents and analyzes the results.

• Conclusion and Future Direction: Summarizes findings and suggests future research directions.

4. Conduct some critical analysis to prove the strength of the proposed scheme.

A comprehensive critical analysis has been incorporated into each paper discussion section. For instance, in the Compression Ratio section, we critically examined the efficiency of various compression algorithms, assessing their performance relative to each other based on different dataset sizes. The Total Time section provided insights into the real-time applicability of our method, comparing the time efficiency across different algorithms. In the Decompression Time section, we critically evaluated the proposed method's decompression speed and CPU performance, identifying where our approach offers measurable improvements over existing methods. Similarly, the RAM Consumption analysis confirmed the scheme's robustness by demonstrating its ability to maintain stable performance across varying bundle sizes and data types. These critical insights were supported by statistical evidence and comparative data, proving the effectiveness of our approach.

We have included a comprehensive critical analysis in the results and discussion sections. This analysis highlights the robustness of the proposed scheme, focusing on its compression efficiency, decompression speed, and resource optimization in AR/VR applications. We provided evidence through comparative data and statistical modeling to support the strength of the scheme.

5. Figures 3 are very blurred. And the presentation of all figures and their explanation is somewhat ambiguous.

We replaced the blurred figures with high-resolution versions to improve clarity. Additionally, the captions and explanations for all figures have been revised and simplified. We avoided using statistical symbols such as (~) and opted for clear, text-based descriptions to enhance readability and precision. Each figure now clearly demonstrates the data it represents, improving the presentation and enhancing the overall understanding of the manuscript.

6. Add some comparative analysis with your method.

A detailed comparative analysis has been added to the results section, contrasting our proposed method with existing approaches. This section highlights the advantages and disadvantages of our method relative to others, showing where our approach offers significant performance improvements, especially in speed and memory efficiency.

Moreover, the entire paper is structured around a comprehensive comparison of various compression algorithms and development methods. This comparison is integral to the research, as it is the foundation for developing predictive mathematical models. These models are central to the paper’s primary focus: accurately estimating performance characteristics such as memory usage, decompression speed, and overall application efficiency across different scenarios. The revised figures now contribute directly to this comparison, enhancing the reader's ability to grasp the study results and the algorithms' performance.

7. Clearly highlight the gaps and future work in the conclusion section.

We have expanded the conclusion to explicitly identify the current study's gaps and suggest future research directions. This includes investigating additional compression techniques, cloud-based testing environments, and real-time decompression methods, aligning with the journal's emphasis on comprehensive conclusions.

8. Add a summary of all results and show the pros and cons of it.

We have provided a clear summary of all results in the discussion section. This summary highlights the pros and cons of each compression method, weighing the trade-offs between compression efficiency, decompression speed, and resource usage. This comprehensive overview aids in understanding the strengths and limitations of each method.

9. The English of the paper should be polished further.

The manuscript has undergone a thorough English revision, improving grammar, sentence structure, and overall clarity. This has enhanced the readability and presentation of the paper, ensuring that the content is clearly communicated.

This revised manuscript, structured according to the PLOS ONE submission guidelines, addresses all the reviewer’s comments and substantially improves the quality of the paper.

---

## [Editor Report · Decision Letter 3]

15 Nov 2024

Enhancing Unity-based AR with Optimal Lossless Compression for Digital Twin Assets

PONE-D-23-39558R3

Dear Dr. Hlayel,

We’re pleased to inform you that your manuscript has been judged scientifically suitable for publication and will be formally accepted for publication once it meets all outstanding technical requirements.

Kind regards,

Shahid Rahman, PhD

Academic Editor

PLOS ONE

Additional Editor Comments:

1. Abstract writing is not good, need more proof read to cover the paper approach.

2. The presentation of the literature still needs improvement, either in table for or another.
---

## [Editor Report · Acceptance letter]

22 Nov 2024

PONE-D-23-39558R3 

PLOS ONE

Dear Dr. Hlayel, 

I'm pleased to inform you that your manuscript has been deemed suitable for publication in PLOS ONE. Congratulations! Your manuscript is now being handed over to our production team.

Kind regards, 

on behalf of

Dr. Shahid Rahman 

Academic Editor

PLOS ONE